# High-resolution visualization of H3 variants during replication reveals their controlled recycling

Camille Clément[1,2], Guillermo A. Orsi[1,2], Alberto Gatto[1,2], Ekaterina Boyarchuk [1,2], Audrey Forest[1,2], Bassam Hajj[3], Judith Miné-Hattab[2,4], Mickaël Garnier[2,4], Zachary A. Gurard-Levin[1,2,5], Jean-Pierre Quivy[1,2] & Geneviève Almouzni[1,2]

DNA replication is a challenge for the faithful transmission of parental information to daughter cells, as both DNA and chromatin organization must be duplicated. Replication stress further complicates the safeguard of epigenome integrity. Here, we investigate the transmission of the histone variants H3.3 and H3.1 during replication. We follow their distribution relative to replication timing, first in the genome and, second, in 3D using super-resolution microscopy. We find that H3.3 and H3.1 mark early- and late-replicating chromatin, respectively. In the nucleus, H3.3 forms domains, which decrease in density throughout replication, while H3.1 domains increase in density. Hydroxyurea impairs local recycling of parental histones at replication sites. Similarly, depleting the histone chaperone ASF1 affects recycling, leading to an impaired histone variant landscape. We discuss how faithful transmission of histone variants involves ASF1 and can be impacted by replication stress, with ensuing consequences for cell fate and tumorigenesis.

[1] Institut Curie, PSL Research University, CNRS, UMR3664, Equipe Labellisée Ligue contre le Cancer, F-75005 Paris, France. [2] Sorbonne Universités, UPMC Univ Paris 06, CNRS, UMR3664, F-75005 Paris, France. [3] Institut Curie, PSL Research University, CNRS, UMR168, Laboratoire Physico-Chimie, F-75005 Paris, France. [4] Institut Curie, PSL Research University, CNRS, UMR3664, F-75005 Paris, France. [5] SAMDI Tech, Inc., Chicago, IL 60657, USA. Correspondence and requests for materials should be addressed to G.A. (email: genevieve.almouzni@curie.fr)

The genome is partitioned into chromatin domains marked by distinct histone variants and their post-translational modifications[1–3]. Cellular identity profiling has emerged on this basis. How this identity is maintained or changed throughout cell division is central to propagate a cell lineage or change cell fate[4]. Chromatin organization undergoes a major challenge during DNA replication. While nucleosomes ahead of the replication fork are disrupted, the corresponding parental histones along with their modifications are recycled on newly synthesized DNA. This process ensures the transmission of parental histone variants with their post-translational modifications[5]. In parallel, de novo deposition of new histones provides a complement to maintain nucleosomal density. While histone deposition occurs rapidly after passage of the fork, restoration of nucleosome positioning and histone post-translational modification profiles takes more time[6,7]. Therefore, it is key to explore how the timing and spatial orchestration of these events participate in maintaining or changing the epigenetic landscape. Notably, replication itself is constantly challenged, and replication stress—caused by secondary DNA structures, DNA damage, nucleotide pool imbalance or mutations in replication proteins—can have short- or long-term consequences for epigenomic stability[8]. In certain cases, this can perturb repressive as well as active histone marks, leading to changes in gene expression patterns. Highlighting the potential impact of this phenomenon, replication stress has often been observed in cancer cells, at early stages of their transformation[8,9].

To date, we have learnt much concerning de novo deposition of new histone variants via pathways involving dedicated histone chaperones[10]. Yet, how parental histone variants are handled to be recycled locally in either normal or stressed conditions remains unclear. To gain understanding into these questions, one must consider (i) their distribution in the genome relative to replication timing, (ii) their 3D spatial configuration in the nucleus relative to replication timing and (iii) the factors involved in their recycling at replication sites in normal and stressed conditions.

The main supply of new histones during replication is provided by increased expression of the replicative histones H3, H4, H2A and H2B[1]. For histone H3, the replicative variants are H3.1 and H3.2[1,2]. In contrast, the H3.3 variant, constitutively expressed, is available throughout all phases of the cell cycle[1,2] and can replace H3.1 at genomic sites undergoing active nucleosome turnover. Consequently, H3.3 is enriched at gene bodies and DNA regulatory elements, reflecting a close association with transcriptional activity, while heterochromatin territories including pericentromeres, transposons and telomeres can also contain this variant[11,12]. Key histone chaperones are involved in de novo deposition of specific histone variants and guide their specific enrichment profiles in the genome[13,14]. The histone chaperone chromatin assembly factor-1 (CAF-1) is specifically associated with H3.1[13] and is key for its deposition coupled to DNA synthesis[15–18], favored through its interaction with proliferating cell nuclear antigen (PCNA)[19,20]. Throughout the cell cycle, the replacement H3 variant H3.3 is deposited in a DNA synthesis-independent manner by a complex comprising the histone chaperone histone regulator A (HIRA)[12,13,21], or by the histone chaperone death-associated protein (DAXX)[14]. Finally, the H3–H4 chaperone anti-silencing function 1 (ASF1)[22] associates with both H3.1 and H3.3 and has been implicated in their storage as well as histone hand-over for de novo deposition, working in concert with CAF-1 or HIRA respectively[23–27].

In mammals, ASF1 exists as two paralogs, ASF1a and ASF1b[28]. In mice, loss of ASF1a is embryonic lethal, while ASF1b deficiency leads to viability but impaired fertility[29], indicating that the two paralogs are not redundant during development[28]. In human

cells, co-depletion of both ASF1a and ASF1b impairs replication fork progression[30,31]. Increasing evidence supports the view that ASF1 could be critical in parental histone recycling. ASF1 forms a complex with the MCM2 subunit of the MCM replicative helicase via a histone H3–H4 bridge[31–33]. Proximity ligation assays enabled the visualization of a fraction of ASF1 at active replisomes[33]. Considering that a recent mass spectrometry study could not reveal the association of ASF1 with the active replicative helicase[34], ASF1 would likely bind transiently to unload the MCM from parental histones and deliver them to the other side of the replication fork. While structural studies show that ASF1 interacts with an H3–H4 dimer[35], parental H3–H4 dimers rarely mix with newly synthesized dimers[36]. In the current view, two ASF1 molecules are required to interact with both H3–H4 dimers while an additional step ensures the reassembly of the original tetramer. Importantly, upon hydroxyurea treatment, replication arrest is concomitant with an accumulation of ASF1 loaded with histones carrying post-translational modifications characteristic of parental histones[31,37]. These might be redeposited at unscheduled sites upon replication restart[37]. Although it remains unclear how replication stress affects the spatial distribution of parental histones, the importance of a tight control of histone recycling emerges as a central mechanism for the integrity of the epigenome[8]. The current model places ASF1 in a key position to recycle parental H3.3 and H3.1, yet its exact role and its contribution to maintaining a histone variant landscape need to be elucidated.

Genome-wide analyses provide a view of how histone variants distribute in the genome. However, their genomic distribution relative to replication timing has never been investigated, contrasting with the extensive characterization of genomic replication patterns directly at the level of DNA[38,39]. Furthermore, advances in super-resolution microscopy have enabled considerable progress in the three-dimensional (3D) visualization of either replication sites[40–42]—allowing the quantification of the number and size of single replicons in human and mouse cells[40]—or histones. Indeed, recent studies visualized histones H2B in different cell types[43,44] and showed that their heterogeneous distribution and density correlate with the pluripotency state[44]. Furthermore, high-resolution visualization of histone post-translational modifications enabled the classification of stem cell states[45], while imaging of the underlying DNA led to a quantitative description of the compaction levels of different chromatin domains[46]. Although the use of histone marks has traditionally enabled the classification of distinct chromatin states[47,48], histone variants are lacking in this picture.

In this work, we exploit a dual approach to study histone variants relative to replication timing with a genome-wide analysis and in 3D using super-resolution microscopy. We combine two-color stochastic optical reconstruction microscopy (STORM)[49,50]—to visualize the histone variants H3.3 and H3.1 relative to replicated DNA—with the SNAP system[12,51,52]—to label specifically global or parental histones. With this assay, we first visualize H3.3 and H3.1 globally at an unprecedented resolution. While both H3.3 and H3.1 cluster in space, we find a distinct spatial configuration for these variants. H3.3 forms spatial domains whose size is not affected by the cell cycle, but whose density decreases throughout S phase. Unlike H3.3, H3.1 forms domains whose size and density are cell cycle dependent. We then specifically probe the recycling of parental histone variants. We perturb histone recycling by targeting (i) DNA itself at the replication fork by inducing replication stress and (ii) histone localization by depleting ASF1. First, we find that replication stress, induced by hydroxyurea, prevents the recycling of parental H3.1 within replicated DNA and impairs its spatial distribution around replicated DNA. Importantly, ASF1 depletion also affects

the recycling of parental H3.1 and H3.3 at replication sites both in terms of quantity and spatial distribution. Most remarkably, upon ASF1 depletion, we observe a change in the spatial distribution of both H3.1 and H3.3 in late S phase, but only of H3.1 in early S phase. Considering the longstanding importance of replication timing for distinct functional events, we discuss the implications of our findings for the maintenance of epigenetic states.

## Results

**Genome-wide analysis of the H3 variant landscape**. We first aimed to assess the genome-wide distribution of the H3 variants H3.3 and H3.1 relative to replication timing. For this, we performed chromatin immunoprecipitation sequencing (ChIP-Seq) to retrieve H3.3 and H3.1 nucleosomes in asynchronous HeLa cells stably expressing tagged H3.3 or H3.1. For each variant, we analyzed the input-normalized coverage at consecutive windows of 10 kb, using the $\log_2$ ratio as a proxy for enrichment or depletion in a given region. Consistent with their dynamics, global H3.3 and H3.1 showed inverse genome-wide enrichment profiles (Fig. 1a). H3.1 tended to accumulate in broad, megabase-sized chromosomal domains that displayed a proportional H3.3 depletion. Conversely, H3.3-rich regions were narrower and exhibited low H3.1 abundance relative to input.

To investigate the occupancy of both variants in relation to replication timing, we included in our analysis Repli-Seq data from Dellino et al.[38], in which replication sites were mapped in six different cell populations (sorted based on 4',6-diamidino-2-phenylindole (DAPI) content) corresponding to consecutive S-phase fractions ($S_1$ to $S_6$, ranging from early to late replicating). We assigned each of the 10 kb windows to an S-phase fraction (chosen as the fraction with the highest mean coverage of bromodeoxyuridine (BrdU) for that window). We evaluated the distribution of H3.3 and H3.1 in each S-phase fraction (Fig. 1b, c). We found that (i) H3.3 is mainly enriched at early-replicating regions and depleted at late-replicating ones and (ii) its occupancy anti-correlates with the timing of replication. Conversely, H3.1 is more uniformly distributed and changes less markedly with replication timing. Late-replicating regions are nonetheless more enriched in H3.1 compared to early-replicating regions (with the exception of the $S_1$ fraction). Genomic regions associated with the $S_6$ fraction show the largest difference between the two variants.

Early-replicating regions tend to coincide with transcriptionally active, gene-rich domains, which are expected to be enriched in H3.3. To examine whether transcription alone could account for the association between histone variants and replication timing, we then integrated nascent RNA-Seq data from Liang et al.[53] in our ChIP-Seq and Repli-Seq analysis. We classified the genomic windows into four categories based on the transcriptional activity (low, mid, high) or the absence of measurable activity (none). We then compared, for each expression category, H3.3 or H3.1 occupancy to replication timing, as measured by BrdU incorporation in the $S_1$ fraction compared to the average incorporation in all S-phase fractions. As expected, H3.1 did not correlate with transcriptional activity (Supplementary Figure 1B) and the relation between H3.1 and replication timing was independent of transcription (Fig. 1d, e). In agreement with previous studies, transcriptional activity associated with early replication and with higher H3.3 occupancy (Supplementary Figure 1A). Interestingly, however, the association between H3.3 and replication timing proved to be independent of transcriptional activity (Fig. 1d, e). Therefore, transcriptional activity alone cannot explain the relationship between H3.3 occupancy and replication timing.

Overall, our approach combining ChIP-Seq, Repli-Seq and nascent RNA-Seq data (i) provides a distribution of H3.3 and

H3.1 in the genome; (ii) shows that H3.3 is enriched in early-replicating chromatin, while H3.1 is enriched in late-replicating chromatin; and (iii) shows that differences in transcription do not fully account for the opposite patterns of H3 variants. How do these findings translate into 3D spatial distribution and nuclear geography? Are they valid at the level of single cells? What are the dynamics of H3.3 and H3.1 that establish and maintain this histone variant landscape, in particular in S phase? To address these questions, we developed an assay to visualize histone variants and regions of replicated DNA using super-resolution microscopy.

**STORM assay to visualize H3 variants and replication sites**. We combined histone monitoring with DNA synthesis labeling. For histones, we exploited two cell lines previously characterized in our laboratory that stably express SNAP-tagged H3.3 or H3.1[12]. The versatility of the SNAP-tag labeling system enables to monitor in vivo global, parental or new histones[12,51,52] (Fig. 2a and Supplementary Figure 7A). For DNA synthesis, we used 5-ethynyl-2'-deoxyuridine (EdU) labeling (later coupled with the fluorophore Alexa 647) and distinguished cells outside S phase, in early S phase and in mid/late S phase based on their typical patterns (Fig. 2b, left). Of note, EdU only labels regions that are undergoing replication during the time of the EdU pulse. Therefore, EdU-negative regions comprise both previously replicated and unreplicated regions. Here, to achieve high spatial resolution and perform a quantitative analysis, we used 3D STORM, which allows the detection of single molecules[49,50]. We first performed a pulse experiment with EdU labeling and successfully reconstituted STORM images for global H3.3 and H3.1 in cells outside of S phase, in early S phase and in mid/late S phase (Fig. 2b, right, see also Figs. 3b and 4b).

We first analyzed the histone signal. Consistent with confocal images, the STORM images showed a broad distribution of H3.1 and H3.3 throughout the nucleus, in cells outside of S phase, in early S phase and in mid/late S phase (Figs. 2b, 3b), while reaching a resolution in the range of 40 nm. At this resolution, we observed that H3.3 and H3.1 distributed heterogeneously in the nucleus, forming groups of detections—reminiscent of those found for histone H2B[44]. To further analyze this signal in the distinct phases, we used a density-based clustering algorithm (DBSCAN) (Supplementary Figure 2A). This clustering method is based on the identification of regions with higher density for H3.3 and H3.1, which we designated as conglomerates (Supplementary Figure 3A and 4A). For each identified histone conglomerate, we measured two parameters, volume and density, which we calculated as the proportion of detections in the nucleus per volume and per conglomerate. We represented the distribution of these parameters for all conglomerates, in cells outside S phase, in early S and in mid/late S phase (see below).

We next examined the EdU signal. For cells outside of S phase, as expected, the EdU signal was low compared to cells in S phase reflecting background noise and possibly some EdU incorporation due to limited DNA damage (Fig. 2b, right). For cells in S phase, we again applied the DBSCAN algorithm to define EdU clusters representative of regions of newly replicated DNA. Here, based on a recent super-resolution microscopy study of replication sites[40], we chose experimental conditions yielding EdU clusters corresponding to several replicons, ensuring that we had enough histone detections within these clusters for quantitative analysis. In our 600 nm sections, we defined an average of respectively 94 and 82 regions of replicated DNA in early and mid/late S-phase cells in the H3.3 cell line, and 122 and 69 in H3.1 (Supplementary Figure 2B), with volumes in the range of $1.5.10^7$ nm$^3$, corresponding to a 300 nm diameter

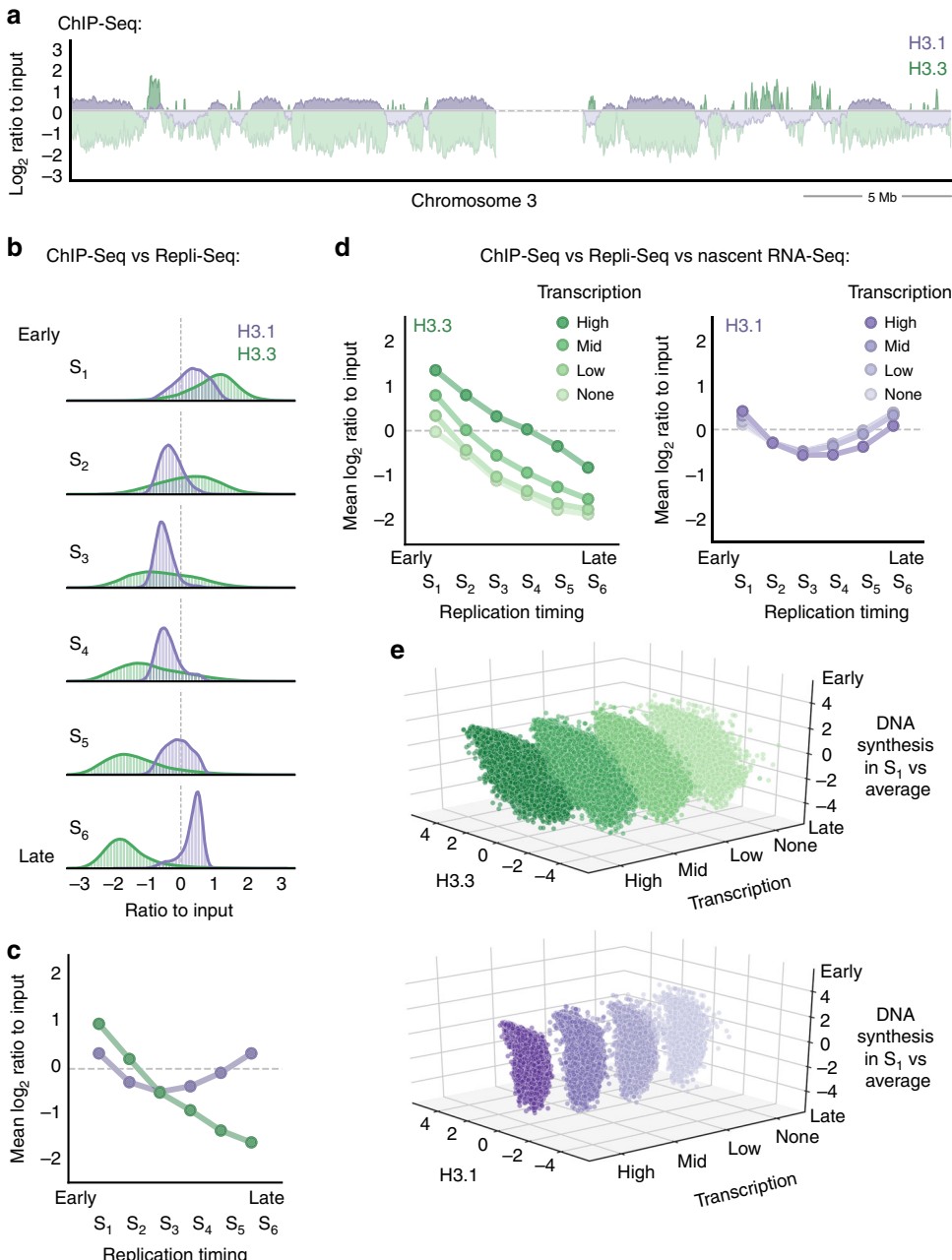

**Fig. 1** Global H3.3 and H3.1 genome-wide occupancy relative to replication timing and transcriptional activity. **a** H3.3 and H3.1 genomic coverage normalized to input (chromosome 3 centromeric band±15 Mb, *x*-axis). The *y*-axis shows the log₂ ratio between the mean per-base number of reads from H3.3 (green) and H3.1 (purple) and their respective input, at consecutive 10 kb bins (smoothed over 5 non-zero bins). Enriched regions (i.e., log₂ ratio ≥ 0) are highlighted in darker colors. **b** H3.3 and H3.1 occupancy by replication timing: the panels show the histogram and Gaussian kernel density corresponding to the log₂ ratio to input for H3.3 (green) and H3.1 (purple) at 10 kb regions ranked by replication timing from early to late (S-phase fraction with the highest mean coverage in Repli-Seq data from Dellino et al.[38]). **c** Mean values corresponding to **b**. **d** H3.3 and H3.1 occupancy by replication timing at increasing levels of nascent transcription: the mean log₂ ratio to input for H3.3 (green, left) and H3.1 (purple, right) at 10 kb regions ranked by replication timing (as in **b**) and transcription percentile (based on nascent RNA-Seq data from Liang et al.[53]). The color gradient represents increasing levels of nascent transcription computed from the log₂-transformed mean coverage: absence of measurable transcription (none), lower 10th percentile (low), 10th to 90th percentile (mid) or upper 10th percentile (high). **e** H3.3 and H3.1 correlation with early DNA synthesis at increasing transcription levels. H3.3 (top) and H3.1 (bottom) log₂ ratio to input (*x*-axis) against S₁ coverage normalized to the average over all fractions (log₂ ratio, *z*-axis) at 10 kb regions ranked by transcriptional status (*y*-axis). The rank reflects the level of nascent transcription as described above

sphere (Supplementary Figure 2C). For an entire nucleus, this would approximately translate into 1000 and 900 replicated regions in early and mid/late S phase, respectively, for H3.3, and 1400 and 800 regions for H3.1, corresponding to about 5 replicons each[40].

**H3.3 conglomerates display cell cycle-dependent density.** Using our STORM-SNAP assay, we first aimed to monitor how the global distribution of H3.3 evolved throughout S phase (Fig. 3a, b). As a first step, we analyzed all H3.3 conglomerates. H3.3 conglomerates had volumes in the range of $4 \times 10^5$ nm³ (90 nm

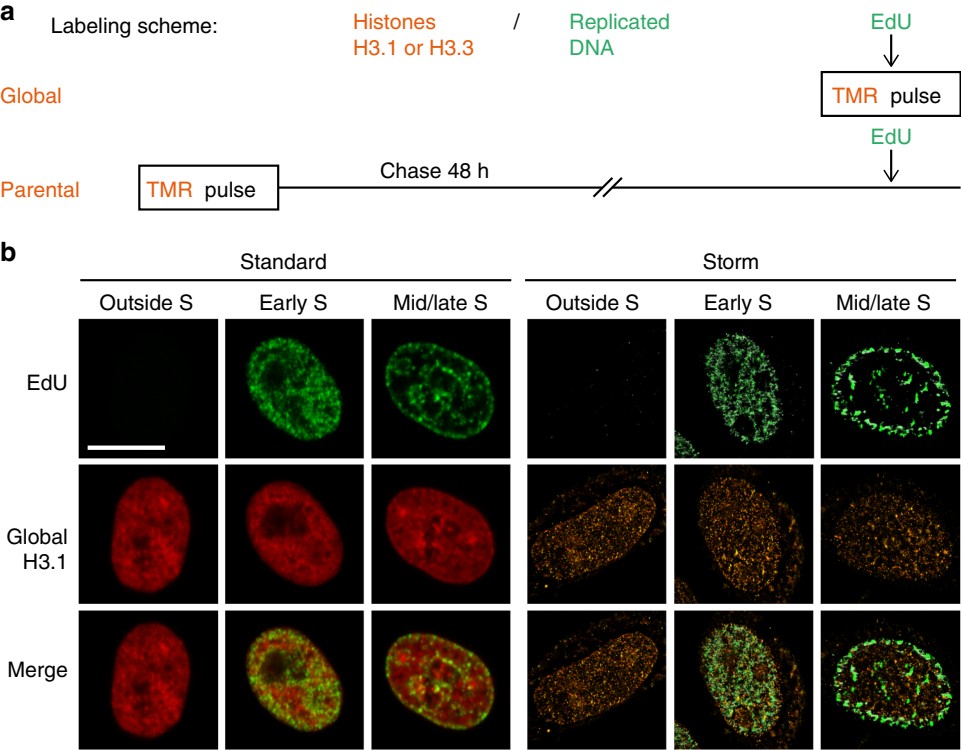

**Fig. 2** Tracking histone H3 variants with STORM microscopy. **a** Labeling scheme using H3.3- or H3.1-SNAP to follow global (top) or parental (bottom) histones. A pulse using the fluorophore TMR (orange) labels SNAP-tagged H3.3 or H3.1. EdU incorporation at the end of the assay allows the detection of replicated DNA (green). This EdU labeling is carried out either during the TMR pulse to compare global H3 distribution with patches of DNA synthesis or after a chase period that allows synthesis and deposition of new unlabeled H3.3- or H3.1-SNAP. The latter enables the localization of 48h-old parental histones with new patches of DNA synthesis. In all cases, we eliminate soluble histones by Triton extraction prior to fixation in order to analyze chromatin-bound H3.3 or H3.1 fractions. **b** Left panels: confocal images of global H3.1 (TMR, red) and replicated DNA (EdU, green) at different S-phase stages and outside S phase. Cells outside S phase are EdU negative, early S phase shows patterns broadly labeling the nucleus with the exception of the nucleoli and mid/late S phase shows patterns with clear enrichment at the nuclear periphery and around nucleoli. Right panels: STORM images of global H3.1 (TMR, orange) and replicated DNA (EdU, green) in cells outside S phase, early S phase and mid/late S phase. We used the ViSP software to render STORM images. Scale bars represent 10 μm

diameter). This compares to H2B regions as described by Ricci et al.[44] (in the range of 80 nm). The distribution of these volumes was stable throughout S phase and outside S phase (i.e.. distributions displayed variability but peaked at identical values and were unrelated to cell cycle stage) (Fig. 3c, top, see also Supplementary Figure 4C). In contrast, the density of H3.3 conglomerates evolved throughout S phase, decreasing when progressing from early to mid/late S phase (Fig. 3c, bottom, see the peak shift −33%, where "+" and "−" refer to increase and decrease, respectively). Cells outside of S phase displayed an intermediate density distribution. Of note, our assay includes a pre-extraction step to eliminate soluble histones. Although we cannot exclude that this treatment could potentially affect our observations, this is necessary to monitor nucleosomal/chromatin-bound histones. Furthermore, to verify that the changes observed in our analysis did not arise from a bias due to the clustering approach, we applied an independent partitioning method (Voronoi tessellation) to our data as described for the nucleoporin protein TPR[54]. We partitioned the signal into polygons such that each polygon contains a single detection, while their size is inversely proportional to local density (Supplementary Figure 3B). The distribution of polygon sizes confirmed that early S-phase cells had more high-density regions than mid/late S-phase cells (Supplementary Figure 3C).

As a second step, we conducted the analysis for H3.3 conglomerates near EdU clusters (under 200 nm from cluster center of gravity), thereby comparing conglomerates in early- and

late-replicating chromatin at sites of DNA synthesis (Fig. 3d, scheme). As in the previous analysis, the volumes of H3.3 conglomerates remained similar (Fig. 3d, top), and their density decreased when progressing from early- to late-replicating regions (Fig. 3d, bottom, see the peak shift −21%, and summary in Fig. 3e).

**H3.1 conglomerates change in volume and density in S phase.** Similarly, we exploited our assay and analytical method to monitor global H3.1 throughout S phase (Fig. 4a, b). H3.1 conglomerates had volumes in the range of 3.5 to $4.5 \times 10^5$ nm$^3$. When analyzing H3.1 conglomerate volumes in the whole nucleus, these were slightly larger in early S compared to outside S phase (+15% peak shift) (Fig. 4c, top), in the same size range as the H3.3 conglomerates described above. Indeed, when we directly compared the volumes of H3.1 and H3.3 conglomerates, H3.1 volumes were slightly larger than H3.3 in early S phase (+16% peak shift), while it was the opposite for outside S and mid/late S phases (−11 and −18% peak shifts) (Supplementary Figure 4B). As these distribution shifts were modest, we looked more specifically at individual cells and refined our observation: H3.1 conglomerates seemed larger in early S than outside S phase, with intermediary volumes for mid/late phase (Supplementary Figure 4C). The volume of H3.1 conglomerates within EdU sites followed the same trend between early- and late-replicating chromatin (−13% peak shift) (Fig. 4d, top), indicating that H3.1

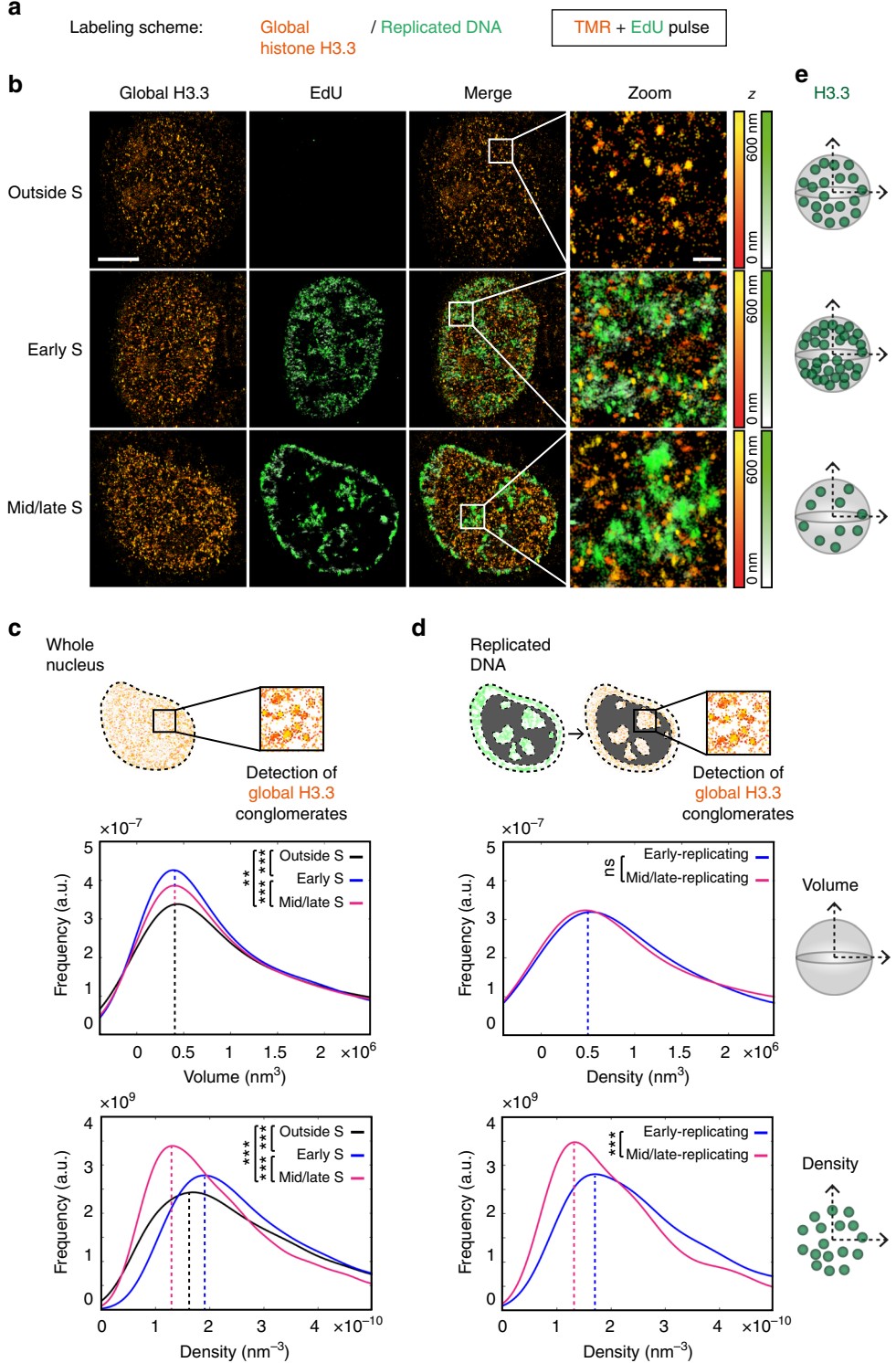

**Fig. 3** Global distribution of H3.3 throughout S phase using STORM assay. **a** Labeling scheme using H3.3-SNAP to follow global H3.3 as described in Fig. 2.
**b** Representative STORM images of global H3.3 (TMR, orange) and replicated DNA (EdU, green) in HeLa H3.3-SNAP. EdU labeling, as described in Fig. 2,
allows the selection of cells outside S phase, in early S phase and mid/late S phase. The color gradient corresponds to the z range. Scale bars represent
5 μm. Insets represent enlarged images of selected area where scale bars correspond to 600 nm. **c** For the H3.3-enriched areas—defined as conglomerates
—in the whole nucleus, the plots show the distribution of volume (top) or density (bottom) of H3.3 conglomerates in cells outside S phase (black), in early
S phase (blue) and in mid/late S phase (magenta). For **c**, **d**, N = 11, 8 and 10 cells for outside S phase, early S phase and mid/late S phase respectively. The
p values (using Mann–Whitney–Wilcoxon test): ***p ≤ 0.001; **p ≤ 0.01; ns not significant. **d** For H3.3 conglomerates in regions of replicated DNA, the
plots show the distribution of volume (top) or density (bottom) of H3.3 conglomerates in cells in early S phase (blue) and in mid/late S phase (magenta).
**e** Scheme summarizing the changes in volume and density of H3.3 conglomerates outside S phase, in early S phase and in mid/late S phase

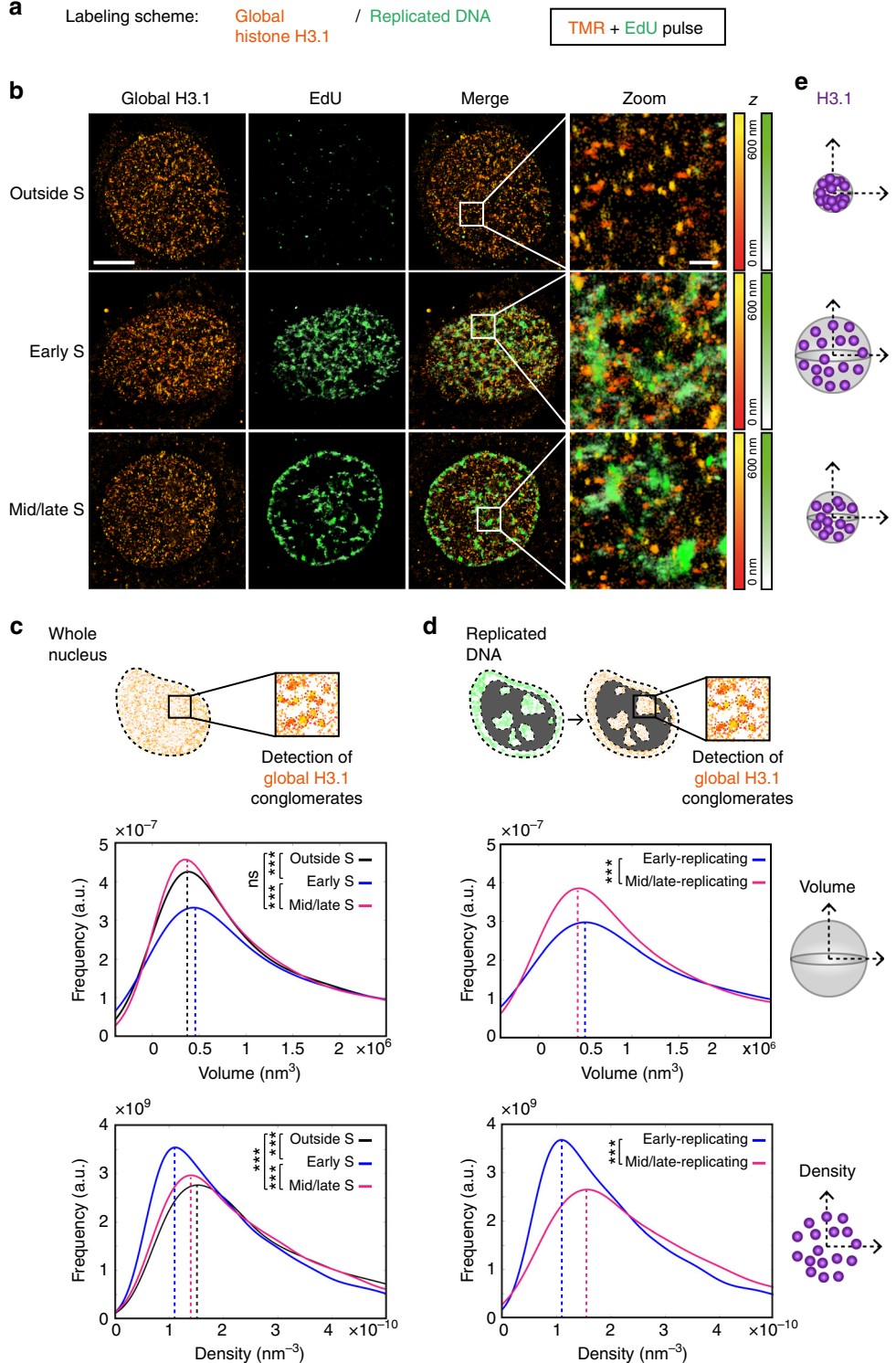

**Fig. 4** Global distribution of H3.1 throughout S phase using STORM assay. **a** Labeling scheme using H3.1-SNAP to follow global H3.1 as described in Fig. 2. **b** Representative STORM images of global H3.1 (TMR, orange) and replicated DNA (EdU, green) in HeLa H3.1-SNAP. EdU labeling, as described in Fig. 2, allows the selection of cells outside S phase, in early S phase and mid/late S phase. The color gradient corresponds to the z range. Scale bars represent 5 μm. Insets represent enlarged images of selected area where scale bars correspond to 600 nm. **c** For the H3.1-enriched areas—defined as conglomerates—in the whole nucleus, the plots show the distribution of volume (top) or density (bottom) of H3.1 conglomerates in cells outside S phase (black), in early S phase (blue) and in mid/late S phase (magenta). For **c**, **d**, N = 9, 10 and 13 cells for outside S phase, early S phase and mid/late S phase respectively. The p values (using Mann–Whitney–Wilcoxon test): ***$p \leq 0.001$; ns not significant. **d** For H3.1 conglomerates in regions of replicated DNA, the plots show the distribution of volume (top) or density (bottom) of H3.1 conglomerates in cells in early S phase (blue) and in mid/late S phase (magenta). **e** Scheme summarizing the changes in volume and density of H3.1 conglomerates outside S phase, in early S phase and in mid/late S phase

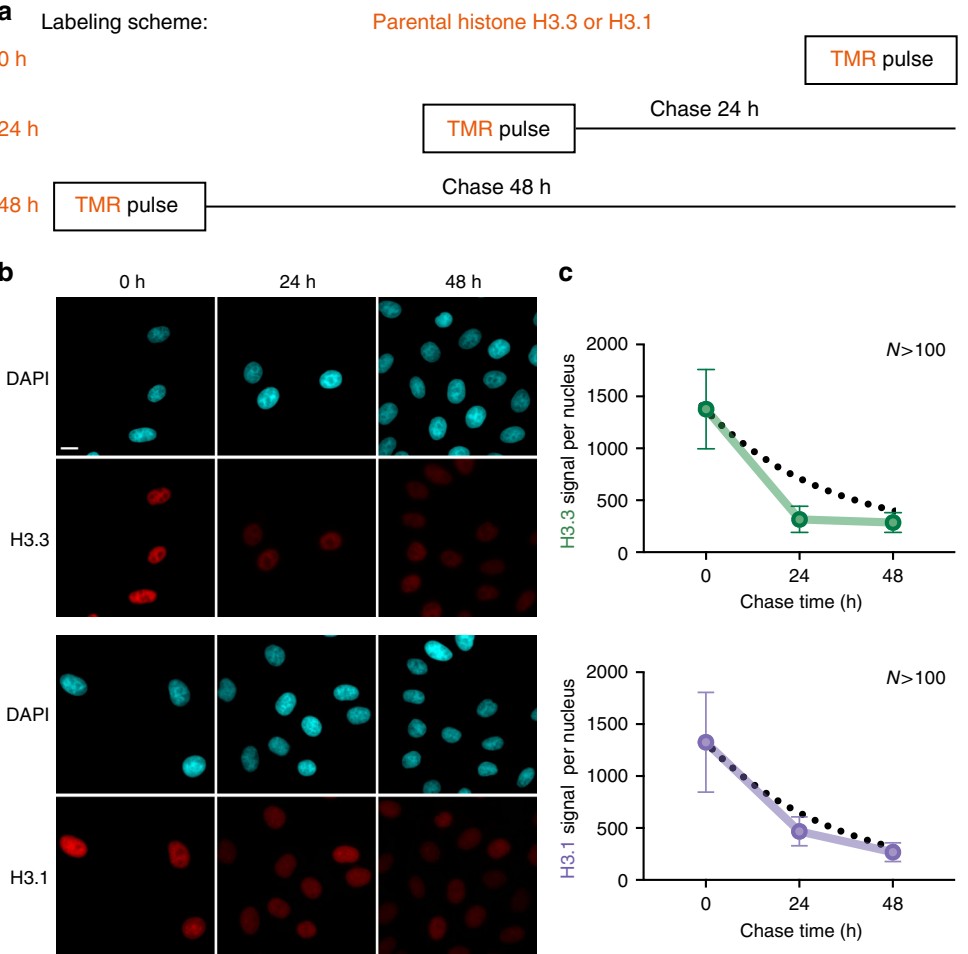

**Fig. 5** Tracking parental H3.3 and H3.1 **a** Labeling scheme using H3.3- or H3.1-SNAP to follow parental H3.3 or H3.1 as described in Fig. 2. We used three different chase times: 0, 24 and 48 h (equivalent to tracking global, 24 h-old and 48 h-old histones). **b** Standard epifluorescence images of histones H3.3 or H3.1 after a chase of 0, 24 and 48 h. DAPI stains nuclei. Scale bars represent 10 μm. **c** The plots show the quantification of the fluorescence for H3.3 or H3.1 after different chase times. More than 100 nuclei were quantified. Error bars represent standard deviation. Dashed lines correspond to how the signal would decrease by cell division-dependent dilution only, determined using the growth rate of each cell line HeLa H3.3- and H3.1-SNAP

conglomerates adopt larger volumes when coinciding with early-replicating chromatin.

We then measured the density of H3.1 conglomerates and found an increase in mid/late compared to early S phase (+26% peak shift) (Fig. 4c, bottom). Of note, the normalization by the total number of detections in the nucleus (Supplementary Figure 4A, right) may in part contribute to these changes. When comparing small surfaces in Voronoi tessellation analysis, we found that mid/late S-phase cells had smaller surfaces than early S-phase cells (Supplementary Figure 4D), confirming our clustering results. The density of H3.1 conglomerates near EdU sites was also increased in late- compared to early-replicating chromatin (+44% peak shift in mid/late compared to early) (Fig. 4d, bottom, and summary in Fig. 4e).

We next investigated how histone variants are maintained at sites of DNA synthesis.

**Monitoring parental histone recycling using the SNAP system**. Our knowledge concerning how histone variants are recycled or discarded at sites of DNA replication remains very limited. To address this question, we labeled specifically parental histone variants using the SNAP system (Fig. 5a, b). We used different chase times (0, 24 and 48 h, corresponding to 0, 1 or 2 cell cycles)

and measured the fluorescence intensity in whole nuclei using epifluorescence microscopy in order to monitor the decrease in the histone signal over several cell divisions (Fig. 5c and Supplementary Figure 5). If parental histones were lost exclusively by S-phase dilution, we would expect an exponential decay, i.e., 50% loss every cell cycle. Instead, we found that both H3.1 and H3.3 are lost at higher rates, consistent with additional replication-independent histone turnover. The increased loss of parental H3.3 compared to H3.1 from 0 to 24 h is in line with our previous result that H3.3 marks early-replicating regions with higher turnover. We focused on a 48 h chase for the following experiments for two reasons: (i) in our experimental conditions, it proved the minimum amount of time required for efficient siRNA depletion (see below), and (ii) a 48 h chase ensured that we specifically examined parental histones without bias linked to cell cycle variation, for both variants H3.1 and H3.3.

**Hydroxyurea impairs local recycling of parental histones**. We next used our STORM assay to first monitor parental histones in a context where we expected local parental histone recycling to be affected. For this, we treated HeLa SNAP H3.1 cells with hydroxyurea (HU) for 30 min before fixation to uncouple helicase progression from DNA synthesis (Fig. 6a). As expected,

HU treatment caused reduced EdU signal (consistent with impaired replication) and the appearance of the single-stranded DNA-binding protein RPA (replication protein A) at replication sites[31] (Supplementary Figure 6A). At the level of total nuclear fluorescence, we detected a slightly higher retention of parental H3.1 in the HU condition compared to the control, consistent with general replication arrest (Supplementary Figure 6B).

Because of the physical uncoupling between parental DNA unwinding and new DNA synthesis, parental histones disrupted ahead of the fork may not be recycled immediately after. To visualize this, we monitored parental H3.1 at EdU sites in HU-treated cells by STORM (Fig. 6b). With low and more disperse signal for parental histones in STORM images compared to global histones, our clustering approach was inappropriate. Instead, we adopted a different strategy: we directly counted the number of parental H3.1 detections in replicated regions (Fig. 6c). We normalized to the EdU signal to account for the fact that HU treatment blocks replication, as well as to the

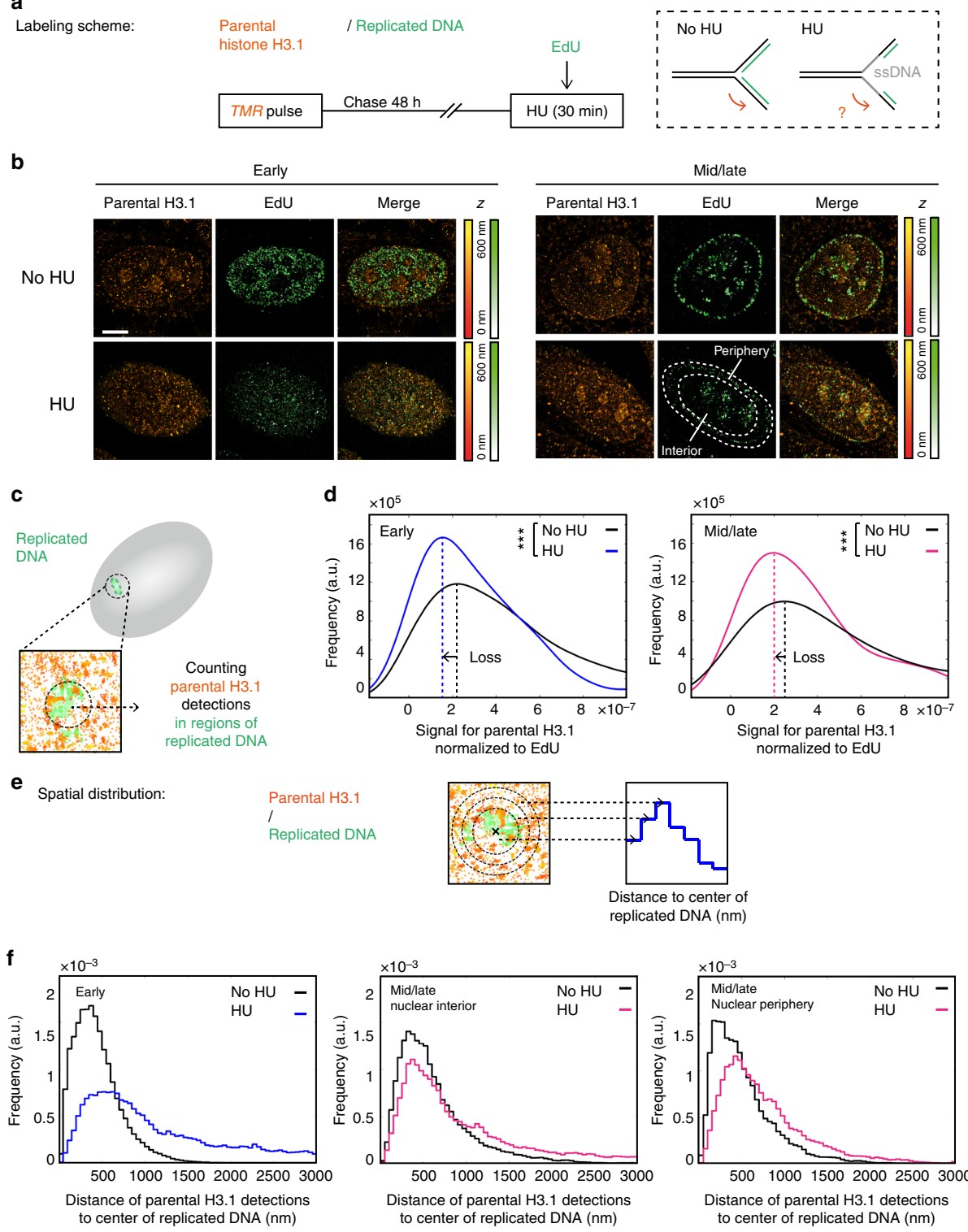

total number of H3.1 detections in the nucleus. As predicted, both in early and mid/late S phases, the amount of parental H3.1 decreased significantly in the HU condition compared to the control (−31% and −21% peak shift) (Fig. 6d). This suggests that local perturbation of DNA at the replication fork inflicted by HU treatment impairs the local recycling of parental histones, leading to a loss of parental histones in regions of replicated DNA.

**Hydroxyurea affects parental histone spatial distribution**. We next asked whether local loss of parental histones had further consequences for their spatial distributions around replication sites. To this end, we developed a method to describe the spatial distribution of histone detections (signal A) around replicated DNA (signal B) (histones vs EdU). We measured the number of histone or EdU detections in concentric regions centered at the gravity center of EdU clusters (Fig. 6e). To validate this approach, we measured the spatial distribution of newly synthesized H3.1 (quench-chase-pulse labeling) vs EdU, using data obtained after performing a quench-chase-pulse experiment (Supplementary Figure 7A, B). Unlike parental H3.3 and H3.1 signal located in the entire nucleus, newly synthesized H3.1 exhibits clear enrichment at EdU-labeled sites, as we previously described[15], providing an ideal context to measure the distance between these two signals. We found that the distance of maximum enrichment between new H3.1 and replication sites centers was 200, 150 and 250 nm in early S phase, mid/late S phase in the interior and mid/late S phase at the periphery (Supplementary Figure 7C, D). To validate these estimates with an independent approach, we adapted a function (termed m function) from a spatial economics study[55]. This function measures the enrichment between two signals, based on the distances between detections of signal A vs detections of signal B. We tested this on simulated data (Supplementary Figure 7E(i)), and then applied it to the new H3.1 data (Supplementary Figure 7E(ii)). This analysis provided a similar estimate of the distance between new H3.1 and EdU as our approach, supporting its validity.

Next, we compared DNA that had been replicated at different times, aiming to detect expected changes in spatial distribution. We performed either a single EdU pulse or an EdU pulse followed by a 30 min chase (Supplementary Figure 8A). Using the DBSCAN clustering analysis, we detected larger replicated regions in the EdU pulse chase than in the EdU pulse (Supplementary Figure 8C). This increase suggested that during the chase period, the organization of the replicated region evolved as a sign of chromatin maturation. We studied the spatial distribution of EdU vs EdU in each experiment, and detected a change towards higher distances in the EdU pulse chase

compared to the EdU pulse (Supplementary Figure 8B), consistent with the clustering result. We concluded that the changes detected in the spatial distribution indeed reflected biological changes, and that earlier-replicated DNA spatially spread further from the center of EdU clusters compared to more recently replicated DNA.

We then applied our method to study the effect of HU treatment on the spatial distribution of parental H3.1 (revealed by pulse-chase labeling) relative to EdU clusters. Upon HU treatment, we observed a clear change in the spatial distribution of parental H3.1 in both early and mid/late S phases (both at the nuclear periphery and interior) (Fig. 6f). In all cases, we found that parental H3.1 redistributed at increased distances from replicated DNA following HU treatment, with H3.1 remoteness increased at a scale of hundreds of nanometers.

Overall, our results show that replication stress upon HU treatment not only leads to local loss of histone recycling, but also their unscheduled redistribution at distant nuclear loci, with a potential impact on the epigenomic landscape.

**ASF1 depletion affects recycling of parental H3.3 and H3.1**. We next investigated which factors recycled histones during DNA replication. As ASF1 was a prime candidate, we performed pulse-chase experiments and downregulated ASF1 (both ASF1a and ASF1b) using small interfering RNA (siRNA) (Supplementary Figure 9A). As previously reported, ASF1 depletion led to slower cell cycle and abnormally shaped nuclei[56]. To check the general effect at a larger scale, we first performed this experiment in cells synchronized with a double thymidine block to verify cell cycle progression, and monitored the dilution of the parental H3.1 signal over two divisions using epifluorescence microscopy (Supplementary Figure 9B). We observed a decrease of the final (48 h) to initial (0 h) signal ratio in the siASF1 condition, suggesting that, over two divisions, ASF1-depleted cells retained parental H3.1 less efficiently than control cells, while we observed no change in the overall nucleosome density (Supplementary Figure 10).

Additionally, we performed immunofluorescence at 48 and 72 h after ASF1 depletion to assess the status of several H3 post-translational modifications typical for nucleosomal H3 in chromatin[57]. We performed this experiment in an asynchronous population and used confocal microscopy to monitor euchromatic H3K4me3 and H3K36me3, and heterochromatic H3K27me3 and H3K9me3 (Supplementary Figure 9C). At this resolution, we did not observe changes in the distribution of the euchromatic marks upon siASF1, although the lack of a clearly defined pattern for these marks might mask subtle changes in distribution. In contrast, we observed a change in the pattern of H3K9me3 and

**Fig. 6** Effect of hydroxyurea treatment on parental H3.1 recycling at replication sites. **a** Labeling scheme using H3.1-SNAP to follow parental H3.1 as described in Fig. 2. In addition, we perform a 30 min hydroxyurea (HU) treatment prior to fixation. The dashed box depicts DNA at the replication fork with or without HU treatment: parental DNA (black), newly synthesized DNA (green) or single-stranded DNA (grey). **b** Representative STORM images of parental H3.1 (TMR, orange) and replicated DNA (EdU, green) in HeLa H3.1-SNAP in early and mid/late S phases. The color gradient corresponds to the z range. Scale bars represent 5 μm. **c** Calculation method for the signal for parental H3.1 normalized to EdU: in regions of replicated DNA, we counted parental H3.1 detections and normalized to the EdU detections and the total parental H3.1 detections in the nucleus. **d** Distribution of the signal for parental H3.1 (right) normalized to EdU in early S phase in control (No HU, black) and HU-treated cells (HU, blue or pink) in early (left) or mid/late (right) S-phase cells. In the No HU condition, $N = 10$ cells for early S and $N = 9$ cells for mid/late S phase. In the HU condition, $N = 8$ cells for early S and $N = 10$ cells for mid/late S phase. The p values (using Mann–Whitney–Wilcoxon test): ***$p \leq 0.001$; ns not significant. **e** Analysis method for the spatial distribution of parental H3.1 relative to replicated DNA. For each region of replicated DNA, we defined 50 nm wide concentric zones centered on the center of gravity of the replicated DNA site. We assigned surrounding detections of parental H3.1 to zones based on their distance to the center. The number of detections counted in each region was normalized to the volume of the corresponding region. **f** Spatial distribution of parental H3.1 relative to replicated DNA. The plots show the distributions of the distances of parental H3.1 to the center of replicated DNA sites for control (black) and HU-treated cells (blue or pink) in early S-phase cells (left), interior of mid/late S-phase cells (middle) and periphery of mid/late S-phase cells (right)

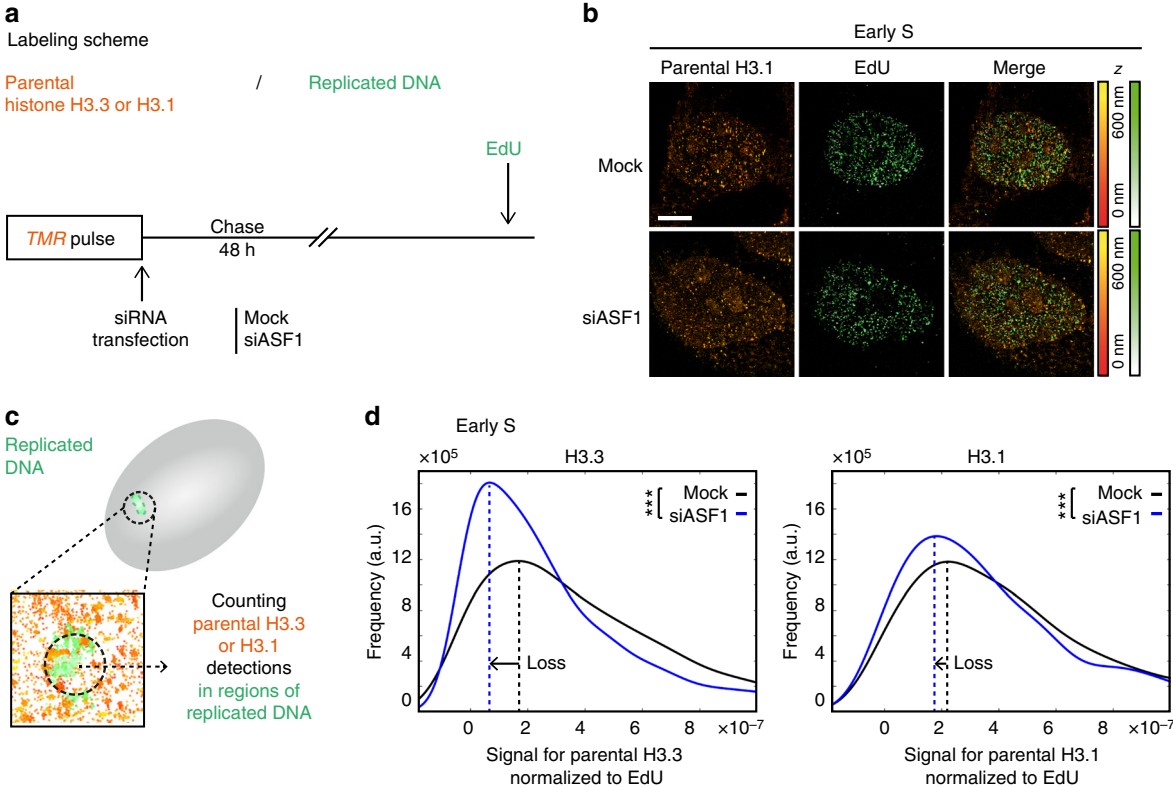

**Fig. 7** Effect of ASF1 depletion on the recycling of parental H3.3 and H3.1 at replicated DNA regions. **a** Labeling scheme using H3.3- or H3.1-SNAP to follow parental H3.3 or H3.1 as described in Fig. 2. In addition, we perform a siRNA transfection (mock or against ASF1) immediately following the TMR pulse labeling histones and prior to the 48 h chase. **b** Representative STORM images of parental H3.1 (TMR, orange) and replicated DNA (EdU, green) in HeLa H3.1-SNAP in early S phase. The color gradient corresponds to the z range. Scale bars represent 5 μm. **c** Calculation method for the signal for parental H3.3/1 normalized to EdU as described in Fig. 6c. **d** The plot shows the distribution of the signal for parental H3.3 (left) or H3.1 (right) normalized to EdU in early S phase in control (mock, black) and ASF1-depleted cells (siASF1, blue). For H3.3, in the mock condition, $N = 11$ cells, and in the siASF1 condition, $N = 10$ cells. For H3.1, in the mock condition, $N = 10$ cells, and in the siASF1 condition, $N = 10$ cells. The p values (using Mann–Whitney–Wilcoxon test): ***$p \leq 0.001$; ns not significant

H3K27me3 in ASF1 depletion conditions, with a more diffuse signal compared to the distribution of these marks at the nuclear and nucleolar periphery in control cells. We then refined our analysis with an increased resolution focusing on H3K36me3 and H3K9me3 using STORM analysis (Supplementary Figure 11A). We found that H3K9me3 formed domains, which decreased in density in the ASF1-depleted condition compared to the control (−46%), with no clear changes in volume (Supplementary Figure 11B). H3K36me3 domains, on the contrary, were unchanged in density, but decreased in volume (−40%) and were more numerous in the ASF1 knockdown (Supplementary Figure 11C, D). Of note, these marks are reestablished on new histones within one cell cycle[7], and they feature different dynamics, and therefore cannot be used as a direct proxy for parental histones over long time periods. Overall, these data suggest that ASF1 depletion affects parental histone recycling at the scale of the entire nucleus, while leading to disorganization of histone post-translational modifications.

To evaluate how ASF1 depletion impacts global H3.1 and H3.3, we performed a pulse labeling experiment following siASF1 knockdown and detected H3.1 and H3.3 conglomerates (Supplementary Figure 12A). For both H3.1 and H3.3 conglomerates, S-phase changes in volumes and densities followed similar trends in siASF1 cells and control cells, although siASF1 cells displayed more variability and less pronounced changes (Supplementary Figure 12B, C and 13). In particular, when focusing on sites of replicated DNA, the density of H3.3 conglomerates in early- vs

late-replicating chromatin was less decreased in the ASF1-depleted condition (−13%) than in the control (−21%), while the density of H3.1 conglomerates increased (+35%) less than the control (+44%). We next aimed to test the effect of ASF1 depletion at sites of replicated DNA in order to directly assess its involvement on parental histone variant recycling.

We focused on parental histones by performing a pulse-chase experiment after depleting ASF1 (Fig. 7a, b), and used STORM imaging to measure the number of parental H3.3 or H3.1 detections at replicated regions (Fig. 7c and Supplementary Figure 14D). As for the HU treatment, we normalized to the EdU signal to account for the fact that ASF1 depletion slows down replication. Strikingly, we found that, in early S phase, the amount of parental H3.3 and H3.1 decreased significantly in the ASF1-depleted condition compared to the control (−60 and −18% peak shift) (Fig. 7d). H3.3 was more affected than H3.1, although we cannot rule out that this reflects a sampling effect rather than a biological difference. In mid/late S phase, we did not observe changes of this magnitude (Supplementary Figure 14A, B, C). However, it is important to note that our data analysis may be underestimating late S-phase defects as we normalized to the total number of detections in the nucleus, which may be affected by early S-phase ASF1-dependent defects.

These results show that ASF1 depletion leads to impaired retention of parental H3.3 and H3.1 at replicated regions most notably in early S phase (Supplementary Figure 17A). The similarity with our result in HU-treated cells suggests that ASF1

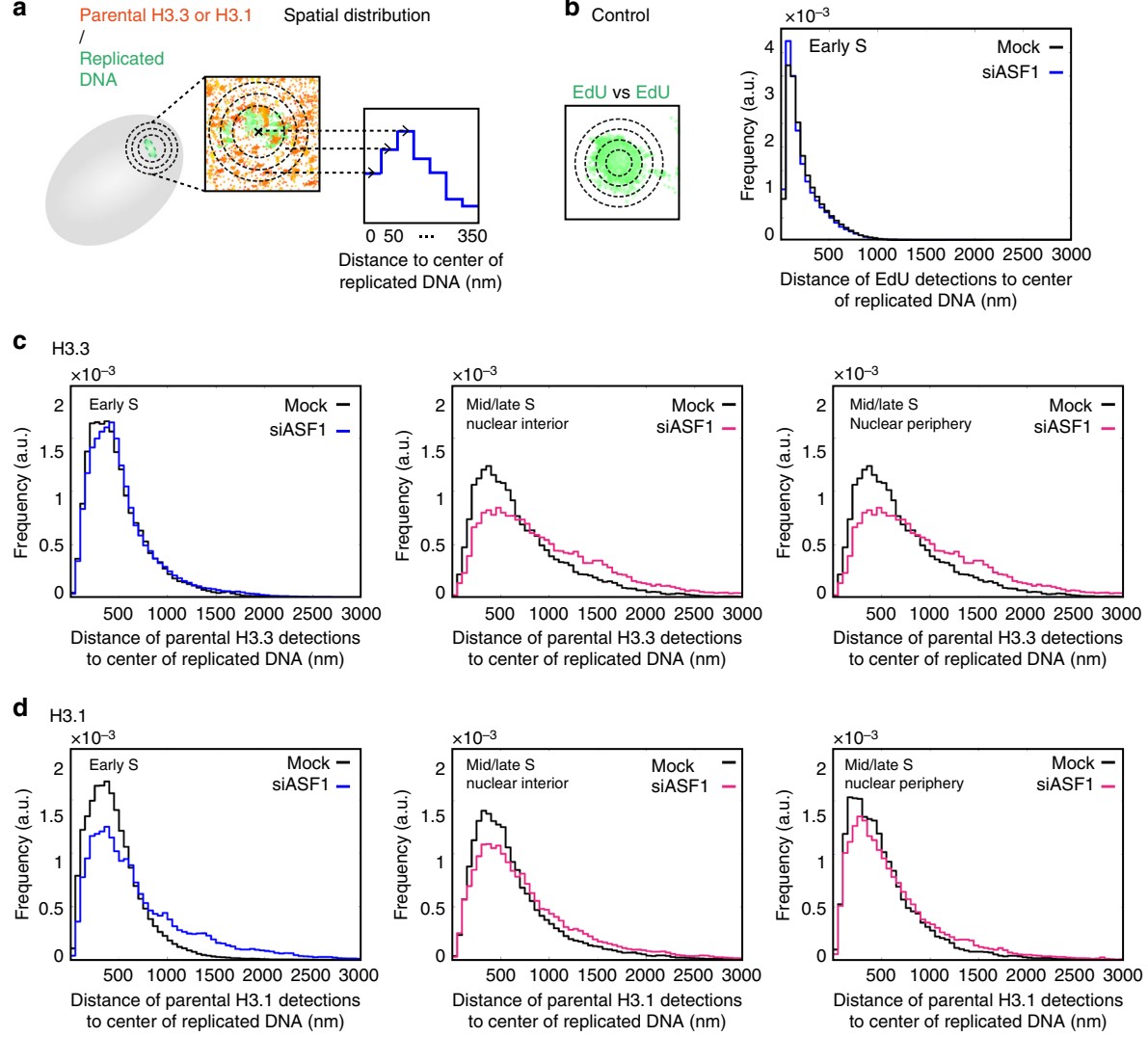

**Fig. 8** Effect of ASF1 depletion on the spatial distribution of parental H3.3 and H3.1 throughout S phase. **a** Analysis method for the spatial distribution of parental H3.3 and H3.1 relative to replicated DNA as described in Fig. 6e. **b** Spatial distribution of EdU relative to replicated DNA. The plot shows the distribution of the distances of EdU detections to the center of replicated DNA sites for control (black) and ASF1-depleted (blue) early S-phase cells. For (**b–d**), the same cells were used as in Fig. 7. **c** Spatial distribution of parental H3.3 relative to replicated DNA. The plots show the distributions of the distances of parental H3.3 to the center of replicated DNA sites for control (black) and ASF1-depleted cells (blue or pink) in early S-phase cells (left), interior of mid/late S-phase cells (middle) and the periphery of mid/late S-phase cells (right). **d** Spatial distribution of parental H3.1 relative to replicated DNA. The plots show the distributions of the distances of parental H3.1 to the center of replicated DNA sites for control (black) and ASF1-depleted cells (blue or pink) in early S-phase cells (left), interior of mid/late S-phase cells (middle) and the periphery of mid/late S-phase cells (right)

depletion affects local histone recycling at the replication fork, likely by uncoupling parental histone transfer from replication fork progression. Of note, ASF1 knockdown does not lead to accumulation of γH2AX[31] (Supplementary Figure 15), suggesting that these effects are not merely a consequence of a DNA damage response.

**ASF1 depletion impairs H3.3 and H3.1 spatial distribution**. The observed decrease in parental H3.3 and H3.1 at replicated regions in early S phase upon ASF1 depletion raised the question of the fate of these lost parental histones. We hypothesized that they could either be degraded or, if recycled, positioned at sites distant from patches of DNA synthesis, which may give rise to changes in their spatial distribution. Therefore, we applied our analytical method to study the impact of ASF1 depletion on the

spatial distribution of parental histones around replicated DNA (Fig. 8a). As an important control, we observed no difference between the siASF1 and the control conditions in the spatial distribution of EdU ("EdU vs EdU" for early S; mid/late S showed no change either) (Fig. 8b and Supplementary Figure 16A, B). This shows that no changes in chromatin compaction or DNA replication speed can account for changes in histone distribution in our analysis. We then investigated whether we could detect differences in histone localization around these sites. We found no difference in the spatial distribution of parental H3.3 vs EdU in early S phase (Fig. 8c, left), despite the overall decrease in amounts of parental H3.3 in replicated regions (see Fig. 7d, left). This indicates that impaired recycling of histones in replicated regions due to ASF1 depletion does not lead to a shift in their spatial distribution. In contrast, when looking at mid/late S-phase cells, both in the nuclear interior and at the nuclear periphery, we

noticed a clear change in the spatial distribution of H3.3 towards farther distances (Fig. 8c, middle and right). Unlike H3.3, parental H3.1 showed a spatial distribution away from replication sites in both early and mid/late S phases (Fig. 8d).

Based on these findings, we conclude that, upon ASF1 depletion-mediated impairment of parental histone recycling, a fraction of parental H3.3 and H3.1 is recycled at distant sites from replication sites, at a scale of hundreds of nanometers away (Supplementary Figure 17B). Intriguingly, parental H3.3 in early S phase, while not properly recycled locally, was not detected to be recycled at distant sites.

## Discussion

Histone variants are a key feature of the epigenetic landscape. In this work, we investigated how their distribution was maintained throughout the cell cycle, and whether replication stress reshuffled their organization. First, we established a genome-wide mapping of H3 variants relative to replication timing. Second, using STORM microscopy and SNAP labeling we further gained a spatial and quantitative view of the relationship between histones and regions of replicated DNA, not only globally, but also with a specific focus on parental histones (Supplementary Figure 18). At a global level, we identified distinct 3D units for H3.3 and H3.1. H3.3 forms units corresponding to early-replicating chromatin with a stable volume throughout the cell cycle. Unlike H3.3, H3.1 forms units that vary both in volume and density during the cell cycle. We can distinguish two categories: (i) in early S phase, large low-density units likely correspond to deposition of new H3.1 in H3.3-associated chromatin, and (ii) at any time during the cell cycle, small high-density units would mark late-replicating chromatin. We then extended our analysis to the fate of parental H3.3 and H3.1 to follow their recycling at an unprecedented level. Using this approach, we could first evaluate how replication stress impairs the recycling of parental H3.1, both in terms of quantity and spatial distribution. Second, we found that ASF1 depletion affected the local recycling of both parental H3.3 and H3.1 at replication sites, but with a distinctive impact on their spatial distribution around replication sites. Therefore, we demonstrate that mislocalization of parental histones ensues in the context of replication stress or histone mismanagement, thereby leading to profound effects on the epigenome.

Genome-wide analysis of H3 variant distribution relative to replication timing revealed an enrichment of H3.3 in early-replicating chromatin, and H3.1 in late-replicating chromatin. Our combined analysis of nascent RNA-Seq with Repli-Seq and our ChIP-Seq data confirmed that H3.3 occupancy correlated with transcriptional activity and early replication timing. Interestingly however, this analysis revealed that transcription alone is not sufficient to explain the correlation between H3.3 occupancy and replication timing. Thus, other mechanisms are likely in place. One possibility is that physical properties of late-replicating chromatin may specifically exclude H3.3. This could either occur by lack of histone turnover or impaired access of the H3.3 deposition machinery. Indeed, late-replicating chromatin coincides which heterochromatic regions that display particular isolating properties, such as phase separation—reported in two recent studies[58,59]—high compaction, as well as the presence of RNAs and chromatin-bound proteins[60]. This could impact H3.3 deposition beyond transcriptional activity alone. While other possibilities cannot be excluded, it will be exciting to explore this avenue in the future using advanced technologies in both physics and genomic studies.

While genomic data allow a precise description of the histone variant landscape, they lack the spatial view of how variants distribute in the nucleus. Our STORM analysis enables the visualization of H3.3 and H3.1 in 3D and reveals that they adopt distinct configurations. H3.3 forms units whose volume is unaffected by the cell cycle. Intriguingly, the volume of H3.3 unit is also independent of their coincidence with early- or late-replicating chromatin, suggesting that this property is independent of whether H3.3 marks euchromatic or heterochromatic sites. Whether this spatial feature relates to intrinsic properties of H3.3 itself is an exciting question to further examine. In addition, H3.3 conglomerate density decreases from early to late S phase. This suggests a dilution of H3.3 in S phase, in line with the fact that the replicative H3.1, but not H3.3, is deposited genome-wide in a DNA synthesis-dependent manner[2]. Outside S phase, the density of H3.3 increases compared to late S phase, likely reflecting a replacement process, with new H3.3 deposition in a DNA synthesis-independent manner at sites where H3.1 had previously been incorporated.

Unlike H3.3, H3.1 forms units with cell cycle-dependent variations in volume and density that we can classify into two categories: (i) large and low-density units present mostly in early S phase that would correspond to H3.3-enriched early-replicating chromatin, and (ii) small and dense units present throughout the whole cell cycle that would mark late-replicating chromatin. The first category diminishes progressively throughout and after S phase. We propose that H3.1 occupies early-replicating chromatin only temporarily, as a means to form new nucleosomes as placeholders coupled to DNA synthesis. Later in S phase and outside S phase, H3.1 would progressively be replaced by H3.3 in early-replicating regions, and the first category of small and dense H3.1 conglomerates—typical of late-replicating chromatin— would become predominant. The changes in density between early- and late-replicating chromatin are also consistent with our genome-wide analysis showing H3.1 enrichment in late-replicating regions.

We conclude that variants enable the definition of distinct units with properties impacted by genomic location and S-phase progression, providing potential novel rules to partition the genome in the nucleus in 3D.

Several studies have shed light on the dynamics of parental histone recycling during replication. Radioactive pulse-chase experiments and electron microscopy studies first showed that parental histones segregated on the two daughter strands of DNA[4]. More recently, using nascent chromatin capture (NCC) in human cells, Alabert et al.[7] detected parental histones with their post-translational marks on newly synthesized DNA and followed the dynamics of reestablishment of these marks on new histones. However, the exact mechanism of recycling has remained unclear, and how perturbing this recycling could impact the spatial distribution of parental histones has never been directly addressed.

To test these mechanisms, we first induced replication stress with hydroxyurea treatment[8]. This causes local effects on DNA— with consequences including fork stalling, checkpoint activation and DNA damage—but its consequences on epigenome maintenance have not been determined. Yet, HU treatment leads to the appearance of single-stranded DNA, which may prevent chromatin assembly at the replication fork, and, supposedly, local recycling of parental histones[37]. We directly tested this latter hypothesis with our STORM assay. Our findings showed unambiguously that HU treatment impaired the recycling of parental H3.1 on replicated DNA. In addition to a loss at replication sites, HU treatment severely impaired the spatial distribution of parental H3.1 in the surrounding region. It would be interesting to monitor the fate of parental histones upon recovery from HU and assess the consequences on the epigenome. Our findings suggest that replication stress may, in some contexts, challenge the integrity of the epigenome and potentially lead to reconfiguration of chromatin territories and unscheduled changes in gene

expression. Indeed, in the context of G-quadruplex-induced stress, impediments on replication were associated with changes in histone mark profiles and gene expression in daughter cells[8]. Interestingly, DNA damage has been found to coordinate the establishment of a protective chromatin environment in regions prone to replication stress, through FACT (facilitates chromatin transcription)-dependent deposition of macroH2A1.2[61], highlighting the importance of dedicated chromatin-mediated mechanisms to face replicative stress. It will be interesting to explore how this relates to H3 variant dynamics. We propose that reshuffling of histone variants upon stress may contribute to epigenomic instability. This may be particularly relevant in the context of cancer, considering the possibility that some oncogenes may induce different forms of replication stress[9,62].

We then investigated the role of factors involved in histone management that could contribute to parental histone recycling. Given that the histone chaperone ASF1 was a prime candidate, we used STORM imaging coupled to the SNAP assay to directly investigate its implication in the recycling and localization dynamics of the parental histone subpopulation. Our results indicated that ASF1 depletion led to (i) a decrease in both parental H3.3 and H3.1 at replicated regions and, importantly, (ii) a change in the spatial redistribution of parental H3.3 and H3.1 relative to replicated regions. When monitoring the effect of ASF1 depletion on conglomerate properties of global H3.3 and H3.1, we detected similar trends as in the control, although less pronounced. In particular, the differences between early- and late-replicating chromatin were decreased. We propose that ASF1 depletion affects variant genomic distribution, while not affecting global S-phase dynamics—such as H3.1 deposition and concomitant H3.3 dilution.

Importantly, the similarity between our results on parental histones following HU treatment and the ASF1 phenotype supports a model in which ASF1 functions directly at the fork to recycle parental histones locally, in line with its capacity to form a complex with the MCM helicase[31–33], potentially in partnership with additional factors to reform nucleosomes. In this model, absence of ASF1 would uncouple the progression of the fork from the transmission of parental histones. As ASF1 depletion, unlike HU treatment, does not trigger replication stress checkpoints—allowing either time for repair or replication arrest as a means to prevent propagation of an abnormal state—this situation is particularly dangerous for the cell, with potential long-term implications for genome and epigenome integrity. Importantly, co-depletion of ASF1a and ASF1b can induce the alternative lengthening of telomeres (ALT) pathway[63]. Telomeres are particularly challenging for replication, as they are prone to fork stalling, formation of secondary DNA structures and DNA damage[64]. In this context, the added stress caused by ASF1 depletion, which prevents parental histone transfer for chromatin reassembly at the fork, might render telomeric regions particularly vulnerable and exposed, triggering in some cases the ALT response. Inversely, it would be interesting to know how ASF1 overexpression may impact the recycling of parental histone variants. In particular, the isoform ASF1b is overexpressed in cancer[56], yet it is unclear whether the two isoforms ASF1a and ASF1b have different roles in recycling H3.3 and H3.1 during proliferation. Taken together, these data further emphasize the importance of histone management in the maintenance of the epigenome.

In the absence of ASF1, we detect changes in the spatial distribution of parental histones. Such changes may reflect that parental histones are recycled at sites distant from their original location. More specifically, this is the case for H3.3 in mid/late S phase and for H3.1 during all S phases, but not for H3.3 in early S phase. ASF1 depletion did not seem to give rise to a spatial

redistribution of the replicated DNA itself, as probed with EdU, suggesting a specific impact on histone localization. How histone variants are handled in this context remains to be elucidated. The most simple hypothesis is that parental histones, if not secured in the vicinity of the replication fork, are treated as new histones and reincorporated further away from their original location by de novo deposition pathways, such as CAF-1 mediated for H3.1 and HIRA or DAXX mediated for H3.3. The distinct fate of parental H3.3 in early S phase may reveal a safeguard mechanism where a fraction of H3.3 would be retargeted locally—while the rest would be degraded—maintaining the spatial distribution of H3.3 in these regions. This could involve, for example, the presence of HIRA and RPA[65]. In this context, it would be interesting to investigate the implication of factors associated with the replication machinery in the recycling of parental histones. Furthermore, we observe changes in the distribution of some histone modifications (H3K9me3 and H3K27me3, H3K36me3). We hypothesize that this may arise from relocation of parental modification-bearing histone variants away from their cognate sites during S phase.

More generally, the presence of DNA damage, transcription machineries[66] or non-nucleosome material[60], and the asymmetry between leading and lagging strand, might interfere with histone recycling and influence the fate of each variant, both in the presence and absence of ASF1. Importantly, ASF1 has been implicated in buffering H3–H4 dimers with nuclear autoantigenic sperm protein (NASP)[23] and in handing them off to CAF-1 and HIRA for de novo deposition[24–27]. It is unclear whether in the absence of ASF1, parental H3–H4 are released as tetramers or as dimers[67], how the histone soluble pool is affected, and how other chaperones can bypass ASF1 function and directly handle tetramers or dimers for their deposition. Notably, parental histones carry post-translational modifications—some of which are more prevalent on specific variants[57]—that may impact their affinity for other factors and, in turn, their fate[2].

Taken together, our observations suggest that ASF1 depletion reshapes the histone variant balance in chromatin during S phase, which may affect transcriptional status. Future work should address the details of the precise mechanism, its regulation and the potential role for other factors in parental histone recycling. Maintaining histone variants at the exact same position during replication encourages a cell to commit to its lineage. Inversely, their loss provides an opportunity to reshape the chromatin landscape. In line with this view, recent studies showed that depletion of CAF-1 facilitated cell reprogramming by pluripotency factors[68,69]. Based on our findings, it is tempting to envisage a similar role for ASF1 in this context. However, considering that ASF1a has been reported to be necessary for maintenance of pluripotency and cellular reprogramming[70] and that ASF1a and ASF1b are involved in multiple pathways[28], the role of ASF1 in differentiation is likely distinct from CAF-1 and awaits further investigation. In particular, in contexts such as the establishment of monoallelic expression—where early replication of the expressed allele coincides with chromatin accessibility[71]—it would be interesting to know whether ASF1 and the distribution of histone variants affect replication timing and, in turn, the differentiation program.

## Methods
**H3.3- and H3.1-SNAP labeling in vivo**. We used cell lines stably expressing H3.3-SNAP-3xHA or H3.1-SNAP- 3xHA in HeLa cells previously used and characterized[12]. These cell lines have been tested negative for mycoplasma contamination. For the pulse experiments, we incubated cells in complete medium containing 2 μM of SNAP-Cell TMR-Star (New England Biolabs) and 10 μM of EdU during 20 min for labeling. We did two quick washes with phosphate-buffered saline (PBS) and then re-incubated the cells in complete medium for 30 min to allow excess SNAP-Cell TMR-Star to diffuse out. We then moved on to extraction

and fixation protocol. For the pulse-chase experiments, we incubated cells with medium containing 2 μM of SNAP-Cell TMR-Star during 20 min for labeling, did two PBS washes and re-incubated the cells in complete medium for 30 min, and then washed twice with PBS again. We incubated the cells in complete medium for a chase period of 48 h. We then washed twice with PBS and re-incubated in complete medium containing 10 μM of EdU for 30 min, before moving on to extraction and fixation protocol. For the quench-chase-pulse experiments, we incubated cells in complete medium containing 10 μM of SNAP-Cell Block (New England Biolabs) to quench SNAP-tag activity, and then performed two PBS washes and 30 min of incubation in complete medium to allow the SNAP-Cell Block to diffuse out. We incubated in complete medium for a 2 h chase period, then performed a pulse step as described above. At least three independent experiments were performed for each condition.

**Extraction and fixation followed by EdU detection.** We performed a pre-extraction of cells prior to fixation for 5 min with 0.5% Triton in CSK buffer (10 mM PIPES (pH 7), 100 mM NaCl, 300 mM sucrose, 3 mM MgCl₂, protease inhibitors), then washed quickly with CSK and performed a 5 min CSK wash. We then fixed cells in 2% paraformaldehyde for 20 min. We blocked cells with bovine serum albumin (BSA; 3% in PBS) before performing Click reaction to reveal the EdU (Click-iT EdU Alexa Fluor 647 imaging kit, Invitrogen). We mounted the coverslips in PBS containing 5 mM of MEA (Mercaptoethylamine, 30070, Sigma) on cavity slides (BR475505, Sigma) and sealed with Twinsil sealing medium (Rotec) before STORM imaging. We changed the mounting buffer between every acquisition.

**siRNA transfection and drug treatment.** In the pulse-chase experiments, we performed a siRNA transfection prior to the pulse chase using Lipofectamine RNAimax (Invitrogen). We used siRNA previously characterized[30,31] against ASF1a (GUGAAGAAUACGAUCAAGUUU) and ASF1b (CAACGAGUACCUC AACCCUUU) at 100 nM concentration (siRNA purchased from Dharmacon). In hydroxyurea experiments, cells were treated with 3 mM HU for 30 min prior to extraction and fixation protocol.

**Micrococcal nuclease sensitivity assay.** One million cells of each indicated condition were collected, washed twice in PBS and nuclei were extracted in 150 mM NaCl, 50 mM Tris-HCl pH 7.5, 2 mM MgCl₂, 0.1% Triton and 0.5% NP-40. Pelleted nuclei were re-suspended in 150 mM NaCl, 50 mM Tris-HCl pH 7.5, 2 mM MgCl₂, 0.1% Triton and 5 mM CaCl₂ buffer, containing 2 units of S7 Micrococcal Nuclease (ThermoFisher Scientific EN0181) and incubated at 37 °C. At each time point, 20% of the sample was collected and mixed with an equal volume of 150 mM NaCl, 50 mM Tris-HCl pH 7.5, 2 mM MgCl₂, 0.1% Triton and 10 mM EGTA to stop the digestion reaction. DNA was extracted, analyzed by electrophoresis in a 1% agarose gel with Sybr Safe dye (Invitrogen) and photo-graphed under ultraviolet light using a ChemiDoc Gel Imaging System (Bio Rad). Images were analyzed with Fiji software to extract density profiles.

**STORM imaging.** We acquired 3D STORM images on a custom setup based on a Nikon iSPT-PALM inverted microscope. We excited Alexa 647—used to label EdU —with a 640 nm laser with a power of 10.8 mW at the sample. We excited TMR—used to label histones—with a 560 nm laser with a power of 41.3 mW at the sample. In addition, both for Alexa 647 and TMR, we used a 405 nm Coherent laser with a power of 18.5 μW at the sample. We imaged the fluorescence from the activated Alexa 647 and TMR molecules with an EM-CCD camera (Ixon Ultra 897 Andor) using a 100×/1.45NA (Nikon) objective. Using this objective, the image pixel size was 160 nm. We used a cylindrical lens (Melles Griot) for 3D[72]. We controlled the microscope with NIS software (Nikon). The number of acquisitions for each experiment is indicated in the figure legends.

We detected the localizations in STORM movies with a custom algorithm as in Sergé et al.[73]. For data visualization, detections were rendered using the ViSP software as an isotropic Gaussian whose full-width half-maximum was 40 nm[74]. We performed z stacks on fluorescent beads (TetraSpeck Microspheres, ThermoFisher) for z calibration. We also used fluorescent beads monitored during STORM image acquisition to correct for sample drift[75] and to align the two signals. When two localizations were detected in consecutive frames within a 50 nm radius, we considered them as one.

**STORM data analysis.** We identified conglomerates of H3.3 and H3.1 or replication foci using the density-based clustering algorithm DBSCAN[76]. DBSCAN uses two input parameters—Eps and MinPts—and determines that a point is in a cluster if at least MinPts points are within a distance of Eps. We used an Eps value of 75 nm and a Minpts of 10 for DBSCAN analysis. For replication foci, we only used clusters with a detection number above a threshold value (100) for further analysis.

To measure the volume of the conglomerates, we used the convex hull function in Matlab. For the density, we calculated the number of detections—normalized by the total number of detections in the nucleus—divided by the volume. We visualized the volume and density as distribution plots using the ksdensity function in Matlab. When looking at H3.3 or H3.1 conglomerates at replication sites, we

selected the ones located under 200 nm from the center of gravity of an EdU cluster; we also tested selecting conglomerates directly situated in an EdU cluster using the convex hull function, which yielded the same results. To study parental H3.3 or H3.1 at replication sites, we calculated the number of H3.3 or H3.1 detections located in the replication site—normalized by the total number of H3.3 or H3.1 detections in the nucleus—and normalized to the EdU signal, accounting for differences in replicative behavior, including between the two cell lines. This was also visualized as a distribution plot using ksdensity. When comparing populations, we used Mann Whitney Wilcoxon test. For comparison between scatter plots, error bars represent standard deviation and we used $t$-test. For the $p$ values: $p > 0.05$ was annotated "ns" (non significant); *$0.01 < p < 0.05$; **$0.001 < p < 0.01$; and ***$p < 0.001$. For the study of spatial distribution, for each replication site, we assigned each surrounding detection of histones (for "histones vs EdU")/EdU (for "EdU vs EdU") to a 50 nm wide concentric region around the center of gravity of the replication site based on its distance to the center of gravity. The number of detections counted in each region was normalized by the volume of the corresponding region and plotted in a bar plot. For the alternative method, we modified the m function from Lang et al.[55]. We considered the distances between all the histone detections vs all the EdU detections and normalized to the distances between detections from two randomly distributed signals. In the plotted graph, when the function is above 1, it indicates attraction, while below 1 indicates repulsion.

**Immunofluorescence and epifluorescence microscopy.** For standard epi-fluorescence imaging of histone post-translational modifications, after blocking with BSA and Click reaction for EdU labeling, coverslips were incubated with primary and secondary antibodies and stained with DAPI. Coverslips were mounted in Vectashield medium. We used an AxioImager Zeiss Z1 microscope with a 63× objective. For confocal images, we used a Confocal Zeiss LSM780, and images were acquired using 63×/1.4NA under Zen blue software (Zeiss Germany). Antibodies were used at the following dilutions: H3K9me3 1:1000 (39765, ActiveMotif), H3K27me3 1:500 (07-449, Millipore), H3K4me3 1:500 (07-473, Millipore), H3K36me3 1:500 (ab9050, Abcam) and γH2AX 1:1000 (05-636, Euromedex).

**H3.3 and H3.1 ChIP and ChIP-seq data analysis.** We performed HA-tag ChIP-seq from the HeLa H3.3-SNAP-3xHA and HeLa H3.1-SNAP-3xHA cell lines as described in Rotem et al.[77]. We used 4 million cells digested in 100 μL with MNase for 8 min at 37 °C (3 units/million cell). We performed HA-ChIP by incubating chromatin (100 μL) supplemented with 500 μL incubation buffer (Tris-HCl 50 mM pH 7.5, NaCl 100 mM, BSA 0.5%, protease inhibitors tablet cocktail, Roche) with anti-HA beads (10 μL) (Roche diagnostics). After overnight incubation on a rotating wheel at 4 °C and following washes, we collected beads in 20 μL TE. We eliminated RNA contaminant by adding 2 μL RNAse A (10 μg/μL) and incubating 30 min at 37 °C. We eluted DNA by adding 2 μL Proteinase K (20 μg/μL), 2.5 μL SDS 2% and incubating 2 h at 37 °C. DNA was then purified with Agencourt AMPure XP Beads (Beckman Coulter) according to the manufacturer's recommendations in 20 μL water. Sequencing libraries (TruSeq ChIP) were prepared with 15 ng of DNA and paired-end sequenced on Illumina HiSeq 2500 at the Institut Curie Next Generation Sequencing (NGS) platform.

Reads were aligned to the human genome version hg19/GRCh37 with Bowtie2[78] (version 2.2.9), run in paired-end mode using the very-sensitive parameter. Genome-wide coverage in bedGraph format was obtained for each alignment using bedtools[79] (version 2.17.0) after sorting and indexing the corresponding BAM file with samtools[80] (version 1.1). Custom Python scripts were used to compute the mean per-base coverage at consecutive 10 kb bins along each chromosome, after normalizing the read counts to the total sequencing depth for each sample. The log₂ ratio to input at non-zero bins was used as a proxy for H3 enrichment.

**Data availability.** All relevant data and analysis code for genome-wide and imaging experiments are available from the authors upon request. Raw sequencing data are available at the European Nucleotide Archive (ENA) under accession number PRJEB27519.

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

## Acknowledgements

This work is dedicated to the memory of our friend and colleague Maxime Dahan. We thank Antoine Coulon and the members of the PIC3i "UNDERSTAND DYNAMIC ARC" for helpful discussions, Shauna Katz and Dominique Ray-Gallet for critical reading of the manuscript, Chloé Guedj, Patricia Le Baccon and the PICT-IBiSA platform for help with the iSPT-PALM microscope. This work was supported by la Ligue Nationale contre le Cancer (Equipe labellisée Ligue), ANR-11-LABX-0044_DEEP and ANR-10-IDEX-0001-02 PSL, ANR-12-BSV5-0022-02 "CHAPINHIB", ANR-14-CE16-0009 "Epicure", ANR-14-CE10-0013 "CELLECTCHIP", EU project 678563 "EPOCH28", ERC-2015-ADG- 694694 "ChromADICT", ANR-16-CE15-0018 "CHRODYT", ANR-16-CE12-0024 "CHIFT", ANR-16-CE11-0028 "REPLICAF" and "Parisian Alliance of Cancer Research Institutes." High-throughput sequencing and Library preparation has been performed by the ICGex NGS platform of the Institut Curie supported by the grants ANR-10-EQPX-03 (Equipex) and ANR-10-INBS-09-08 (France Génomique Consortium) from the Agence Nationale de la Recherche ("Investissements d'Avenir" program), by the Canceropole Ile-de-France and by the SiRIC-Curie program–SiRIC Grant "INCa-DGOS- 4654".

## Author contributions

G.A. supervised the work. G.A. and C.C. conceived the overall strategy and wrote the paper. C.C. performed epifluorescence and STORM experiments and analyzed STORM data. G.A.O. performed epifluorescence, confocal and STORM experiments, analyzed STORM data and wrote corresponding parts in the paper. A.G. analyzed the epigenomics data and wrote corresponding parts of the paper. E.B. performed STORM microscopy experiments. A.F. performed the chromatin immunoprecipitation experiments. B.H. and J.M.-H. built the custom PALM/STORM microscope. M.G. developed tools for image analysis. Z.A.G.-L. conceived and analyzed cell biology experiments. J.-P.Q. designed the chromatin immunoprecipitation experiments and wrote corresponding parts of the paper. Critical reading and discussion of all data involved all authors.

## Additional information

**Competing interests:** The authors declare no competing interests.

