## [Peer Review File · Nature Communications]

Reviewers' comments:

Reviewer #1 (Remarks to the Author):

This manuscript by Almouzni and colleagues describes an innovative approach to dissect histone H3 variant distribution through replication. A link between the ASF1 chaperone and histone recycling is presented. Loss of ASF1 impairs recycling of both H3.1 and H3.3 in a way that resembles the impact of replication stress by HU. Overall, the work is well presented and the data are convincing, albeit at times modest in their effect. While the focus of this manuscript is the characterization of dynamic H3 variant behavior in S phase at unprecedented resolution, mechanistic insight into the underlying processes is limited to ASF1, a known mediator of H3 recycling. Moreover, conclusions drawn from the observations are often speculative, e.g. the proposed replacement of H3.1 with H3.3. No evidence is presented that H3.1 low density conglomerates in early S correspond to H3.3 rich early replicating chromatin (lines 284-300 and discussion, line 488). While this is an interesting and insightful study, several textual and experimental adjustments are needed.

Specific comments:

- 1) It is unclear why the authors decided to use a 48 h chase for their SNAP approach to monitor parental histones, given their claim that this subset likely reflect parental histones incorporated into specialized regions of low turnover (line 324). As such, it would be difficult to draw general conclusions from these experiments. A subset of key analyses should be repeated with a chase of 24 h or less. This would also increase signal intensity and the likelihood to detect conglomerates.
- 2) The notion that the observed histone conglomerate changes in S phase are not limited to local DNA synthesis is intriguing. Can the authors show if similar, global changes are observed upon HU treatment? Is this also observed for parental histones? How do they reconcile this with their replication-coupled recycling model and the statement that local replication perturbation accounts for histone loss at regions of replicated DNA (line 356)?
- 3) Regarding the proposed loss/exchange of H3 variants, how do the authors control for overall nucleosome density?
- 4) Are the HU-induced changes in histone variant density/distribution due to DNA damage signaling?
- 5) Line 302: Doesn't H3.1 density decrease in early S phase?
- 6) Line 387: the expanded EdU size upon chase could also reflect chromatin expansion, which would need to be factored into statements regarding H3 variant density and distance. H3 variants may not be lost/redistributed but rather diluted or spread out through decondensation, similar to findings by Adams et al, Mol Cell 2017.
- 7) Line 552: recent work by Kim et al., Mol Cell 2018 on replication stress-associated epigenome maintenance should be discussed in this context.
- 8) Fig. 2 seems more appropriate for the supplement. Did the authors account for the differences in replicative behavior in H3.1- and H3.3-SNAP cells (Fig S2B, C)?
- 9) Global H3.3 images in Fig 3A do not seem representative of the statement that H3.3 domains don't change in size or density. Foci appear smaller in early S.

Reviewer #2 (Remarks to the Author):

In this manuscript Clement et al. investigate the distribution of histones H3.1 and H3.3 in the genome during unperturbed replication or after damage. They combine a genomic approach with super resolution and find differences in the partition of these histone throughout the genome. They find H3.3 at mainly at the early replicating genome and H3.1 at the late. In 3D space they find that H3.1 forms variable domains but H3.3 forms more stable domains. Interestingly they show that upon depletion of ASF1 chaperon both H3.1 and H3.3 are redistributed unevenly ways from sites of replication.

The study is solid and the observations are interesting. Having said that the manuscript will benefit from simpler description of the super resolution data and their analysis.

It would be also more complete if the superresolution data are combined with histone modifications co stainings for active transcription (RNAPol II or H3K36me3) or silent genome and heterochromatin (H3K9me3). Heterochromatin domains can be further correlated with centromeres or telomeres in wt and DAX or ATRX ko conditions.

Reviewer #3 (Remarks to the Author):

To accurately transmit epigenetic information, cells have to ensure the correct deposition of histone variants during DNA replication. Here, Clément et. al. use a combination of Chip-Seq and single-molecule localization microscopy to address how the histone H3 variants H3.3 and H3.1 are distributed along the genome and how they are propagated during replication.

I believe the data are sound and the manuscript is well written. However, the data should be analyzed more carefully:

1. Distributions of density of conglomerates (e.g. Fig. 3C bottom and entire manuscript)

To calculate the density of a conglomerate, the authors use the number of localizations within a conglomerate normalized to the total number of localizations within a nucleus. I understand that this normalization may be necessary to account for cell-cell variability, however, it can also easily distort the density distributions: An increase in the total number of localizations per nucleus may result in an apparent decrease in density even if the actual density, i.e. the actual number of H3.3/H3.1 per volume within a conglomerate, remains constant. Given that histones are synthesized during the cell cycle, the authors have to verify that their normalization does not falsify their results.

I.e. does the total number of localizations per nucleus change during the cell cycle? If yes, is this change sufficient to explain changes in density distributions?

2. Evaluation of H3.3/H3.1 conglomerates/localizations near replication sites (as first described in line 252 and line 348):

The authors use a sphere of radius 500 nm from the EdU cluster center to identify H3 near replication sites.

- Why did the authors choose a radius of 500 nm?

- As exemplified in the scheme in Fig. 6C, EdU clusters are not necessarily spherical and 500 nm spheres may contain a mixture of H3 associated with replication sites and more distant H3.

Therefore, changes in the shape of replication sites could result in changes in the distributions of apparently associated H3 because a higher (or lower) fraction of distant H3 is included.

Considering that some of the changes observed are small, a better way of identifying replication site associated H3 would be beneficial. E.g. can the authors use DBSCAN of both channels to identify clusters?

3. I find the paragraph "Monitoring parental histones recycling using the SNAP system" highly

confusing:

- Fig. 5C: I am assuming that "H3.3/H3.1 signal" refers to signal per nucleus. Please clarify the axis label and/or figure legend.

- line 322: What do the authors mean by "the non-linearity of the signal decrease"?

Assuming exponential growth conditions and stable paternal histones, the paternal histone fluorescence signal should decrease exponentially (Fig. 5C), or, as described in the text, the percentage of decrease should be constant (i.e. 50% per cell division), but neither would be linear.

- Fig. 5C: Interestingly, the decrease in fluorescence signal is not exponential, and I think this observation deserves more analysis:

a) What is the growth rate of the cells analyzed? I.e. is the deviation from an exponential decrease explained by differences in growth rates?

b) If the experiment is performed under exponential growth conditions, Fig. 5C should be a semi-log plot to highlight any deviations from an exponential decrease.

c) Comparing the decrease in paternal histone fluorescence signal to the growth rate would then allow the authors to assess any potential histone loss that is independent of the dilution by cell division.

4. Spatial distribution of histone detections around regions of replicated DNA (as first described in paragraph 357-402).

- Why did the authors choose to use the method relying on concentric regions rather than the adapted m function that would be independent of EdU cluster shape or chromatin spreading?

According to Fig S6D and Fig. S6E(ii), the m function analysis seems to be more robust.

- If using concentric regions to describe differences in spatial distributions, the authors should verify that EdU cluster size or shape are not affected by any treatment or cell cycle phase.

5. DBSCAN analysis:

- How was the Eps value of 75 nm determined? A figure showing the k-dist graphs as described in ref. 86 might be helpful.

- line 700: Since readers might not be familiar with DBSCAN, the roles of the parameters Eps (maximal distance between points within a cluster) and Minpts (minimal number of points within a cluster) should be stated.

Other points:

- line 239: Fig. S3D does not exist

- line 305-310: The distributions of H3.1 and H3.3 do not seem to be static but are dynamically reorganized. Considering these dynamics, how are the observed distributions affected by the pre-extraction used for sample preparation?

- Fig. 1 and Fig. S1: labeling of the axes should include the fact that "log₂(ratio)" is shown as indicated in the figure legend

- lines 442-444: Can the authors show a plot for total detections per nucleus to demonstrate how the total number of detections in the nucleus is affected by ASF1 depletion? This analysis might also be interesting regarding the fate of parental histones upon ASF1 depletion (recycling or degradation, lines 607-613).

Minor points:

- line 196: maybe include the reference to Fig. 3B for H3.3 localization.

- line 271: ... 4.5×10^5 nm³. Wh

- lines 684, 685, 686, 740, Fig. S4D (color legend): use "." rather than "," as decimal separator.
- Fig. 3B, 4B, 6B, 7B, S6B and S9A: The colors (two channels and z-range) of merged images are hard to interpret. I suggest removing the z-information from merges and a wider/more distinct z-color range for single channel images.
- Fig. S2B: What are the error bars (s.d. or s.e.)? Are the differences observed significant? Maybe also include these errors in the text (line 216).
- ksdensity plots: The y-axes are labelled "Fraction of ... (a.u.)". While I understand that the authors did not want to use the term "density" to avoid confusions, a "fraction" is unit-less and should not be used. Maybe the authors can find a better alternative for density.
- Fig. S4D: It is hard to compare the volumes in this plot. Maybe the authors could include a scatterplot of the peak volumes of individual cells.

Point by point answer to reviewers' comment

Reviewer #1 (Remarks to the Author):

This manuscript by Almouzni and colleagues describes an innovative approach to dissect histone H3 variant distribution through replication. A link between the ASF1 chaperone and histone recycling is presented. Loss of ASF1 impairs recycling of both H3.1 and H3.3 in a way that resembles the impact of replication stress by HU. Overall, the work is well presented and the data are convincing, albeit at times modest in their effect. While the focus of this manuscript is the characterization of dynamic H3 variant behavior in S phase at unprecedented resolution, mechanistic insight into the underlying processes is limited to ASF1, a known mediator of H3 recycling. Moreover, conclusions drawn from the observations are often speculative, e.g. the proposed replacement of H3.1 with H3.3. No evidence is presented that H3.1 low density conglomerates in early S correspond to H3.3 rich early replicating chromatin (lines 284-300 and discussion, line 488). While this is an interesting and insightful study, several textual and experimental adjustments are needed.

We thank the reviewer for appreciating the innovative approach used in our work to dissect H3 variant distribution through replication and for his/her overall positive assessment of our manuscript. To our knowledge this is the first time that dual color analysis is used in STORM to follow histones and DNA synthesis. This is a highly demanding technology and in this first story, we felt that it was important to focus on the technology and analytical tool and apply this to one candidate factor, ASF1, to address the processes.

We surely hope that this will be inspiring to explore other components mechanistically, for example at the level of the helicase (MCM), and/or accessory factors associated with the replication machinery. We have now introduced these aspects for the discussion to pave the way for future research, which are beyond the scope of this manuscript.

We have now also edited our statements to make a clear distinction between facts and speculations. In particular, the most speculative parts have been either removed or placed into the discussion to avoid confusion.

Below, we provide point-by-point answers to the specific concerns.

Specific comments:

1) It is unclear why the authors decided to use a 48 h chase for their SNAP approach to monitor parental histones, given their claim that this subset likely reflect parental histones incorporated into specialized regions of low turnover (line 324). As such, it would be difficult to draw general conclusions from these experiments. A subset of key analyses should be repeated with a chase of 24 h or less. This would also increase signal intensity and the likelihood to detect conglomerates.

Indeed we realize that this choice deserved a clearer explanation. We chose the 48-hour chase for two reasons: (i) in our experimental conditions, it proved the

minimum amount of time required for an efficient siRNA depletion, and (ii) a 48-hour chase ensured that we specifically examined parental histones without bias linked to cell-cycle variations, for both variants H3.1 and H3.3. Indeed, at this time point, the variations between the two variant for parental histones due to cell cycle is negligible. We agree with the reviewer that this was not evident and have now clarified it in the text (line 290).

This timing, however, gives rise to low signal. This does not allow the monitoring of conglomerates in the ASF1 knockdown, as noted by the reviewer.

Therefore, we decided to set up a distinct type of experiment. 48h after siASF1, we performed a single pulse to visualize global H3.1 and H3.3, to guarantee both efficient knockdown and the capacity to detect conglomerates. Importantly, in this case, we monitor both parental and new histones. However, it is critical to note that the fraction of new histones only represents ~5% of the total as measured in Ray-Gallet et al. 2011, thus, we approximate that we have a proxy with this new experiment to estimate the behavior of parental histones after siASF1 treatment. Our findings show that cells depleted in ASF1 featured similar properties as the control when comparing conglomerate density and volume in early vs late S phase cells, yet the variation were systematically reduced (new Supplementary Figure 12 and line 400). In particular, when comparing H3.3 conglomerates in early- vs late-replicating chromatin, their density slightly decreased (-13% peak shift) but not as much as the decrease in density observed in the control condition (-21%). Their volume, as in the control, was unchanged.

Our new results show that (i) in absence of ASF1, the distribution of each variant – in particular in early- vs late-replicating regions - is affected, as if their enrichment at corresponding loci was blurred compared to the control, and (ii) meanwhile, global S phase changes in conglomerate properties still occur in absence of ASF1. The latter is possibly due to unchanged histone variant dynamics throughout S phase (H3.1 deposition/H3.3 dilution) thought to be mediated by other factors, such as CAF-1 for deposition of H3.1 or HIRA and DAXX for deposition of H3.3. These findings strengthen our work on the ASF1 depletion to further document perturbations at the global level for each histone variant. We have now included these data along with our previous analysis.

2) The notion that the observed histone conglomerate changes in S phase are not limited to local DNA synthesis is intriguing. Can the authors show if similar, global changes are observed upon HU treatment? Is this also observed for parental histones? How do they reconcile this with their replication-coupled recycling model and the statement that local replication perturbation accounts for histone loss at regions of replicated DNA (line 356)?

Indeed, the effects we observe in conglomerate properties are not limited to EdU-labeled regions. However, it is important to note that in our experiments, EdU only labels regions that are undergoing replication during the time of the EdU pulse (typically 20 minutes). Therefore, EdU-negative region comprise both previously replicated and un-replicated regions. With this in mind, the global changes in

conglomerate properties could still reflect a consequence of replication, although not limited to EdU-labeled sites. For example, we detect more dense H3.1 conglomerate in late S phase compared to early S phase in the whole nucleus. Indeed, in late S phase, the vast majority of the genome has already been replicated and therefore has incorporated new H3.1 thereby increasing the enrichment of this variant in the whole genome. We now clarified this in the text (line 189).

What happens at EdU-labeled sites and in the rest of the nucleus is indeed linked. As a consequence, replication-coupled recycling of histones at EdU-labeled sites does impact histone distribution in the whole nucleus, as exemplified in our new experiment monitoring conglomerate properties upon ASF1 depletion (new Supplementary Figure 12), as described above.

3) Regarding the proposed loss/exchange of H3 variants, how do the authors control for overall nucleosome density?

We thank the reviewer for raising this point. We have now included an MNase digestion assay to verify that nucleosome density does not change significantly upon ASF1 knockdown (new Supplementary Figure 10).

4) Are the HU-induced changes in histone variant density/distribution due to DNA damage signaling?

We also wondered whether DNA damage signaling could be the reason for the changes observed after HU. Since ASF1 depletion leads to similar observations without detection of any sign of DNA damage - as shown in Groth et al. 2007, and further confirmed in a new supplementary figure panel showing similar phosphorylated H2A.X levels in control and ASF1 knockdown cells (Supplementary Figure 15) - it is unlikely that our findings upon HU treatment are solely a consequence of DNA damage signaling.

5) Line 302: Doesn't H3.1 density decrease in early S phase?

Yes, the reviewer is correct, this was improperly formulated. This sentence has now been removed.

6) Line 387: the expanded EdU size upon chase could also reflect chromatin expansion, which would need to be factored into statements regarding H3 variant density and distance. H3 variants may not be lost/redistributed but rather diluted or spread out through decondensation, similar to findings by Adams et al, Mol Cell 2017.

This point is also raised by Reviewer 3. This possibility was important to consider in light of our previous work (Adam et al, Mol Cell 2017). In our original manuscript, we did in fact consider that H3 distribution could have been due to decondensation rather than redistribution. To ensure that changes in H3 distribution did not

trivially reflect changes in newly-replicated DNA distribution, we included as a control: EdU signal distribution relative to EdU sites, and showed no detectable change in control versus ASF1 condition (Figure 8B). To further document this important point, we have now added a new analysis showing the same result in early and late replicating regions (Supplementary Figure 16A, B). This is now reported in the text (lines 441). We thank the reviewers for helping us to make this point entirely clear.

7) Line 552: recent work by Kim et al., Mol Cell 2018 on replication stress-associated epigenome maintenance should be discussed in this context.

We thank the reviewer for pointing this recent publication that we now discussed, and was appropriate in our manuscript (lines YY). We included that DNA damage has been found to coordinate the establishment of a protective chromatin environment in regions prone to replication stress, through FACT-dependent deposition of macroH2A1.2, highlighting the importance of dedicated chromatin-mediated mechanisms to face replicative stress.

8) Fig. 2 seems more appropriate for the supplement. Did the authors account for the differences in replicative behavior in H3.1- and H3.3-SNAP cells (Fig S2B, C)?

We would prefer to keep Figure 2 in the main figures, as it allows us to better explain the two-color STORM analysis, which is important for the understanding the rest of the manuscript.

Concerning the replication behavior of the two cell lines, in order to draw our interpretations: (i) we always normalize our measurements to EdU signal, therefore controlling for differences in replicative behavior between the two cell lines; (ii) we never directly compare measurements from both cell lines. Therefore differences between cell lines cannot affect our conclusions. We have further underlined this in the text for clarity (lines 729).

9) Global H3.3 images in Fig 3A do not seem representative of the statement that H3.3 domains don't change in size or density. Foci appear smaller in early S.

We agree with the reviewer, we have replaced this with a more representative example (Figure 3).

Reviewer #2 (Remarks to the Author):

In this manuscript Clement et al. investigate the distribution of histones H3.1 and H3.3 in the genome during unperturbed replication or after damage. They combine a genomic approach with super resolution and find differences in the partition of these histone throughout the genome. They find H3.3 at mainly at the early replicating genome and H3.1 at the late. In 3D space they find that H3.1 forms

variable domains but H3.3 forms more stable domains. Interestingly they show that upon depletion of ASF1 chaperon both H3.1 and H3.3 are redistributed unevenly ways from sites of replication.

The study is solid and the observations are interesting. Having said that the manuscript will benefit from simpler description of the super resolution data and their analysis.

We thank the reviewer for his/her overall positive comments on our manuscript. This is novel methodology and it was thus a critical concern for us to make the procedure accessible to the reader. We have made considerable efforts to further clarify these sections. We thus hope that the message remains complete yet more accessible for a general readership.

It would be also more complete if the superresolution data are combined with histone modifications co-stainings for active transcription (RNAPol II or H3K36me3) or silent genome and heterochromatin (H3K9me3).

We thank the reviewer for this useful suggestion. In the original version of the manuscript, we had only included epifluorescence images showing redistribution of some of these marks (H3K36me3, H3K4me3, H3K9me3 and H3K27me3) upon ASF1 knockdown (Supplementary Figure 8). We have now included new super-resolution experiments and data analysis to document the distribution of H3K36me3 and H3K9me3 upon ASF1 knockdown (new Supplementary Figure 11). Consistent with our experiment using epifluorescence microscopy, our new super resolution analysis shows again that H3K9me3 domains decrease in density in the ASF1-depleted condition compared to the control, with no clear changes in volume. Remarkably, H3K36me3 domains, in the knockdown experiment, while unchanged in terms of density, appear more numerous and show a decrease in volume in the ASF1 knockdown. The ability to detect these changes was clearly a major advance thanks to the resolution in STORM, which cannot be achieved with epifluorescence microscopy. These new results strengthen our original conclusions that ASF1 affects the distribution of histone modifications, and analysis of STORM images provided a description of the effect at an unprecedented level.

Heterochromatin domains can be further correlated with centromeres or telomeres in wt and DAX or ATRX ko conditions.

The question as to whether histone variants and modifications get redistributed away from heterochromatin, and in turn whether there are possible consequences for centromeric chromatin was important to consider. To address this issue, we decided to follow the fate the centromeric histone H3 variant CENP-A, as well as the centromeric protein CENP-C and thus performed immunofluorescent staining of both proteins. Upon ASF1 knockdown, we did not observe any significant change at their level (see Reviewer Figure 1, data not included in the paper). Although we cannot exclude minor effect below the resolution of this approach, we cannot at this

stage assign a specific role to ASF1 in handling the centromeric H3 histone. Future work will be needed for a deeper analysis of the dynamics of this variant.

Concerning telomeres and the connection with DAXX/ATRX, while this is clearly a very interesting question, considering the amount of work to carry out the analysis in super resolution as we did it for ASF1, this cannot be part of the same manuscript and it deserves, on its own, a complete study that is thus beyond the scope of our current manuscript.

Reviewer #3 (Remarks to the Author):

To accurately transmit epigenetic information, cells have to ensure the correct deposition of histone variants during DNA replication. Here, Clément et. al. use a combination of Chip-Seq and single-molecule localization microscopy to address how the histone H3 variants H3.3 and H3.1 are distributed along the genome and how they are propagated during replication.

I believe the data are sound and the manuscript is well written. However, the data should be analyzed more carefully:

We thank the reviewer for his/her overall positive assessment of this manuscript and for prompting us to provide more controls for our analysis to deepen our work. Below is a detailed point-by-point answer to the data analysis concerns raised.

1. Distributions of density of conglomerates (e.g. Fig. 3C bottom and entire manuscript)

To calculate the density of a conglomerate, the authors use the number of localizations within a conglomerate normalized to the total number of localizations within a nucleus. I understand that this normalization may be necessary to account for cell-cell variability, however, it can also easily distort the density distributions: An increase in the total number of localizations per nucleus may result in an apparent decrease in density even if the actual density, i.e. the actual number of H3.3/H3.1 per volume within a conglomerate, remains constant. Given that histones are synthesized during the cell cycle, the authors have to verify that their normalization does not falsify their results. I.e. does the total number of localizations per nucleus change during the cell cycle? If yes, is this change sufficient to explain changes in density distributions?

We agree with the reviewer that what we measure and refer to as “density” is not an absolute number but rather the fraction of total detections in each conglomerate. As raised by the reviewer, this is a necessary normalization step to account for cell-to-cell variations. Density is therefore influenced by the total number of detections. We have added the total number of detections (Supplementary Figure 3A (right) and 4A (right)) and have clarified how we define “density” in the text (line 207). As density and total detections are interdependent, we cannot formally separate them, but our

conclusions are reinforced in the case of H3.3. Indeed, the number of total detections changes during the cell cycle in a different manner than density. Thus, these values cannot account for one another, indicating that our approach does reveal relevant changes throughout the cell cycle. For H3.1, we observe similar trends for cell cycle changes in density and total detections. We have highlighted this in the text to clarify that we cannot distinguish the contribution of each (line 270). Yet, we should underline that, when we performed these experiments in the ASF1 knockdown (new Supplementary Figure 12 and line 400), we observed similar trends as in the control for the density with no changes in the total number of detections (new Supplementary Figure 13), arguing that density changes are not only a consequence of total number of detections.

2. Evaluation of H3.3/H3.1 conglomerates/localizations near replication sites (as first described in line 252 and line 348):

The authors use a sphere of radius 500 nm from the EdU cluster center to identify H3 near replication sites.

- Why did the authors choose a radius of 500 nm?

- As exemplified in the scheme in Fig. 6C, EdU clusters are not necessarily spherical and 500 nm spheres may contain a mixture of H3 associated with replication sites and more distant H3. Therefore, changes in the shape of replication sites could result in changes in the distributions of apparently associated H3 because a higher (or lower) fraction of distant H3 is included. Considering that some of the changes observed are small, a better way of identifying replication site associated H3 would be beneficial. E.g. can the authors use DBSCAN of both channels to identify clusters?

Indeed, we have applied DBSCAN to both channels to identify EdU sites already in the original version of the manuscript.

First, the reviewer is however correct in pointing out that the choice of a distance threshold to include histone conglomerates seems arbitrary. Our threshold in the data presented here was actually 200 nm, which we corrected in the text (line 247, 723).

Second, as suggested, we have now identified histone conglomerates in actual replication sites as defined by EdU signal, rather than at this uniform distance. We have found that this gives rise to essentially the same results (Reviewer Figure 2, data not included in the manuscript). This modification to our analysis thus strengthens our original conclusions.

3. I find the paragraph "Monitoring parental histones recycling using the SNAP system" highly confusing:

- Fig. 5C: I am assuming that "H3.3/H3.1 signal" refers to signal per nucleus. Please clarify the axis label and/or figure legend.

This has been clarified.

- line 322: What do the authors mean by “the non-linearity of the signal decrease”? Assuming exponential growth conditions and stable paternal histones, the paternal histone fluorescence signal should decrease exponentially (Fig. 5C), or, as described in the text, the percentage of decrease should be constant (i.e. 50% per cell division), but neither would be linear.

The reviewer is correct, the expected decrease is indeed exponential, we have modified this in the text (line 285).

- Fig. 5C: Interestingly, the decrease in fluorescence signal is not exponential, and I think this observation deserves more analysis:

a) What is the growth rate of the cells analyzed? I.e. is the deviation from an exponential decrease explained by differences in growth rates?

b) If the experiment is performed under exponential growth conditions, Fig. 5C should be a semi-log plot to highlight any deviations from an exponential decrease.

c) Comparing the decrease in paternal histone fluorescence signal to the growth rate would then allow the authors to assess any potential histone loss that is independent of the dilution by cell division.

The reviewer is absolutely correct. The signal decay is not exponential, and we do observe histone loss that is independent of cell division. This observation is expected as other mechanisms lead to histone turnover, such as transcription or DNA repair. To clarify this point we have included the theoretical exponential decrease curve expected based on the growth curves of each cell line and if histone loss was solely due to cell division dilution, highlighting that we do observe additional histone loss (Figure 5C, new Supplementary Figure 5). We have also clarified this section in the text (line 285).

4. Spatial distribution of histone detections around regions of replicated DNA (as first described in paragraph 357-402).

-

Why did the authors choose to use the method relying on concentric regions rather than the adapted m function that would be independent of EdU cluster shape or chromatin spreading? According to Fig S6D and Fig. S6E(ii), the m function analysis seems to be more robust.

We thank the reviewer for raising this point. Indeed, in the control we chose to show, where EdU and new histone signal strongly colocalize (or attract each other at a specific distance), the m function proved quite robust. However, applying the m function in a context where the two signals do not display colocalization/attraction, such as for EdU vs. parental histones, this analysis only detects lack of spatial attraction, but cannot to detect differences between control and knockdown.

- If using concentric regions to describe differences in spatial distributions, the authors should verify that EdU cluster size or shape are not affected by any treatment or cell cycle phase.

We agree with the reviewer that this is an important point, also raised by Reviewer 1. In our original manuscript we included a control for this concern. We plotted the distribution of EdU signal around EdU sites in different conditions to ensure that there was no change in this distribution that could trivially account for the changes we see in histone distribution (Figure 8B). We now included an additional analysis showing these distributions in early but also late EdU clusters (Supplementary Figure 16), and made efforts to highlight this in the text (lines 441).

5. DBSCAN analysis:

- How was the Eps value of 75 nm determined? A figure showing the k-dist graphs as described in ref. 86 might be helpful.- line 700: Since readers might not be familiar with DBSCAN, the roles of the parameters Eps (maximal distance between points within a cluster) and Minpts (minimal number of points within a cluster) should be stated.

We thank the reviewer for raising this point. Indeed, clustering requires a threshold value. We set this threshold by visual comparison of the clusters detected by DBSCAN and the original images. We made an effort to further clarify these points, as also suggested by Reviewer 2 (line 235).

Other points:

- line 239: Fig. S3D does not exist

This has been corrected.

- line 305-310: The distributions of H3.1 and H3.3 do not seem to be static but are dynamically reorganized. Considering these dynamics, how are the observed distributions affected by the pre-extraction used for sample preparation?

Indeed, we apply pre-extraction in sample preparation. Although we cannot exclude that this treatment could potentially affect our observations, it is necessary to eliminate soluble histones and focus on chromatin bound/nucleosomal histones. We now state this reservation for our conclusion (line 235).

- Fig. 1 and Fig. S1: labeling of the axes should include the fact that “log₂(ratio)” is shown as indicated in the figure legend

This has been corrected.

- lines 442-444: Can the authors show a plot for total detections per nucleus to demonstrate how the total number of detections in the nucleus is affected by ASF1 depletion?

This analysis might also be interesting regarding the fate of parental histones upon ASF1 depletion (recycling or degradation, lines 607-613).

In order to monitor how the total signal is affected by ASF1 depletion, we have included a quantification based on epifluorescence images (Supplementary Figure 9B), showing that, overall, parental histone loss is more pronounced in the knockdown. This value is more robust than absolute total detections in STORM because (i) it measures total nuclear signal rather than subsampling of “blinking” molecules in a thin section; (ii) it includes hundreds of nuclei. We still plotted the total nuclear detection counts as requested (Supplementary Figure 14D), and detected no changes between control and ASF1 knockdown.

Minor points:

- line 196: maybe include the reference to Fig. 3B for H3.3 localization.

This has been amended.

- line 271: ... 4.5×10^5 nm³. When...

This has been amended.

- lines 684, 685, 686, 740, Fig. S4D (color legend): use “.” rather than “,” as decimal separator.

This has been amended.

- Fig. 3B, 4B, 6B, 7B, S6B and S9A: The colors (two channels and z-range) of merged images are hard to interpret. I suggest removing the z-information from merges and a wider/more distinct z-color range for single channel images.

We agree that there is no ideal solution to plot 3D information in a 2D figure. We tried to apply the reviewer’s suggestion but this resulted in an image that visually exaggerated colocalization. We thus decided to keep our current color code.

- Fig. S2B: What are the error bars (s.d. or s.e.)? Are the differences observed significant?

Maybe also include these errors in the text (line 216).

We have included this information.

- ksdensity plots: The y-axes are labelled “Fraction of ... (a.u.)”. While I understand that the authors did not want to use the term “density” to avoid confusions, a

“fraction” is unit-less and should not be used. Maybe the authors can find a better alternative for density.

We agree, we will use the more widely accepted term “frequency”.

- Fig. S4D: It is hard to compare the volumes in this plot. Maybe the authors could include a scatterplot of the peak volumes of individual cells.

We have included this plot (Supplementary Figure 4C). However, as this is only partial information, we have kept our original plot as well.

Reviewer Figure 1

CENPA (red) and CENPC (grey) stainings in control and ASF1-depleted conditions, revealed by immunofluorescence in HeLa H3.1-SNAP cells. DAPI stains nuclei (blue). The images were acquired using an epifluorescence microscope. Scale bars represent 10 μ m.

Reviewer Figure 2

For H3.1 conglomerates in regions of replicated DNA: the plots show the distribution of volume (A) or density (B) of H3.1 conglomerates in cells in early S phase (blue), and in mid/late S phase (magenta). These graphs plot the same data as in Figure 3D, but H3.1 conglomerates were considered in a region of replicated DNA when their center of gravity was located in the convexhull of an EdU cluster (see Reviewer 3 question 2).

REVIEWERS' COMMENTS:

Reviewer #1 (Remarks to the Author):

The authors have addressed my concerns and I recommend publication of the manuscript. One comment with regard to data presentation: to be able to better interpret the newly added ASF1 knockdown data (supplemental Fig 12) and how they relate to the results for parental histone (Fig 7D), it would be helpful to include a comparison with control cells as done in Fig 7D.

Reviewer #2 (Remarks to the Author):

The authors have satisfied all my concerns with their revised manuscript.

Reviewer #3 (Remarks to the Author):

The authors have addressed all my concerns and I fully support publication of this manuscript. However, the authors should consider some minor changes:

1. Labeling of the y-axes in ksdensity plots ("Frequency") should include units.
[Minor point: Fig. 3C, bottom left: "Frequenct" should read "Frequency"]
2. Fig. S3A/S4A, right and Fig. S13: The number of frames used for the quantification of localization events per nucleus should be stated.

Point by point response to reviewers:

Reviewer #1 (Remarks to the Author):

The authors have addressed my concerns and I recommend publication of the manuscript. One comment with regard to data presentation: to be able to better interpret the newly added ASF1 knockdown data (supplemental Fig 12) and how they relate to the results for parental histone (Fig 7D), it would be helpful to include a comparison with control cells as done in Fig 7D.

We added these graphs in Supplementary Figure 12.

Reviewer #2 (Remarks to the Author):

Th authors have satisfied all my concerns with their revised manuscript.

Reviewer #3 (Remarks to the Author):

The authors have addressed all my concerns and I fully support publication of this manuscript.

However, the authors should consider some minor changes:

1. Labeling of the y-axes in ksdensity plots ("Frequency") should include units. [Minor point: Fig. 3C, bottom left: "Frequent" should read "Frequency"]
2. Fig. S3A/S4A, right and Fig. S13: The number of frames used for the quantification of localization events per nucleus should be stated.

We made the requested changes in the figures.